# Unsupervised Behavior Extraction via Random Intent Priors

**Hao Hu**[1][*]**, Yiqin Yang**[2][*]**, Jianing Ye**[1]**, Ziqing Mai**[1]**, Chongjie Zhang**[3]
[1]Institute for Interdisciplinary Information Sciences, Tsinghua University
[2]Department of Automation, Tsinghua University
[3]Department of Computer Science & Engineering, Washington University in St. Louis
{huh22, yangyiqi19, yejn21, maizq21}@mails.tsinghua.edu.cn
chongjie@wustl.edu

## Abstract

Reward-free data is abundant and contains rich prior knowledge of human behaviors, but it is not well exploited by offline reinforcement learning (RL) algorithms. In this paper, we propose **UBER**, an unsupervised approach to extract useful behaviors from offline reward-free datasets via diversified rewards. UBER assigns different pseudo-rewards sampled from a given prior distribution to different agents to extract a diverse set of behaviors, and reuse them as candidate policies to facilitate the learning of new tasks. Perhaps surprisingly, we show that rewards generated from random neural networks are sufficient to extract diverse and useful behaviors, some even close to expert ones. We provide both empirical and theoretical evidence to justify the use of random priors for the reward function. Experiments on multiple benchmarks showcase UBER's ability to learn effective and diverse behavior sets that enhance sample efficiency for online RL, outperforming existing baselines. By reducing reliance on human supervision, UBER broadens the applicability of RL to real-world scenarios with abundant reward-free data.

## 1 Introduction

Self-supervised learning has made substantial advances in various areas like computer vision and natural language processing (OpenAI, 2023; Caron et al., 2021; Wang et al., 2020) since it leverages large-scale data without the need of human supervision. Offline reinforcement learning has emerged as a promising framework for learning sequential policies from pre-collected datasets (Kumar et al., 2020; Fujimoto & Gu, 2021; Ma et al., 2021; Yang et al., 2021; Hu et al., 2022), but it is not able to directly take advantage of unsupervised data, as such data lacks reward signal for learning and often comes from different tasks and different scenarios. Nonetheless, reward-free data like experience from human players and corpora of human conversations is abundant while containing rich behavioral information, and incorporating them into reinforcement learning has a huge potential to help improve data efficiency and achieve better generalization.

How can we effectively utilize the behavioral information in unsupervised offline data for rapid online learning? In online settings, Eysenbach et al. (2018); Sharma et al. (2019) investigate the extraction of diverse skills from reward-free online environments by maximizing diversity objectives. Meanwhile, in offline settings, a rich body of literature exists that focuses on leveraging the dynamic information from reward-free datasets (Yu et al., 2022; Hu et al., 2023). However, the former approach is not directly applicable to offline datasets, while the latter approach overlooks the behavioral information in the dataset. Ajay et al. (2020); Singh et al. (2020) employ generative models to extract behaviors

---

[*]Equal contribution.

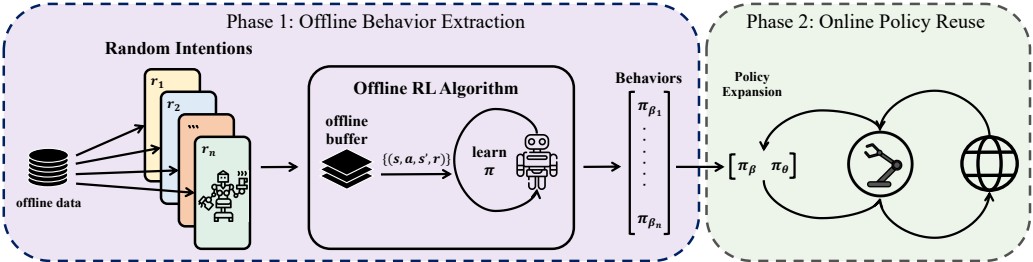

Figure 1: The framework of UBER. The procedure consists of two phases. In the first offline phase, we assign different reward functions to different agent to extract diverse and useful behaviors from the offline dataset. In the second phase, we reuse previous behavior by adding them to the candidate policy set to accelerate online learning for the new task.

in the offline dataset. However, using a behavior-cloning objective is conservative in nature and has a limited ability to go beyond the dataset to enrich diversity.

To bridge this gap, we propose **U**nsupervised **B**ehavior **E**xtraction via **R**andom Intent Priors (UBER), a simple and novel approach that utilizes random intent priors to extract useful behaviors from offline reward-free datasets and reuse them for new tasks, as illustrated in Figure 1. Specifically, our method samples the parameters of the reward function from a prior distribution to generate different intents for different agents. These pseudo-rewards encourage the agent to go beyond the dataset and exhibit diverse behaviors. It also encourages the behavior to be useful since each behavior is the optimal policy to accomplish some given intent. The acquired behaviors are then applied in conjunction with policy expansion (Zhang et al., 2023) or policy reuse (Zhang et al., 2022) techniques to accelerate online task learning. Surprisingly, we observe that random intent prior is sufficient to produce a wide range of useful behaviors, some of which even approach expert performance levels. We provide theoretical justifications and empirical evidence for using random intent priors in Section 4 and 5, respectively.

Our experiments across multiple benchmarks demonstrate UBER's proficiency in learning behavior libraries that enhance sample efficiency for existing DRL algorithms, outperforming current baselines. By diminishing the reliance on human-designed rewards, UBER expands the potential applicability of reinforcement learning to real-world scenarios.

In summary, this paper makes the following contributions: (1) We propose UBER, a novel unsupervised RL approach that samples diverse reward functions from intent priors to extract diverse behaviors from reward-free datasets; (2) We demonstrate both theoretically and empirically that random rewards are sufficient to generate diverse and useful behaviors within offline datasets; (3) Our experiments on various tasks show that UBER outperforms existing unsupervised methods and enhances sample efficiency for existing RL algorithms.

### 1.1 Related Works

**Unsupervised Offline RL.** Yu et al. (2021) considers reusing data from other tasks but assumes an oracle reward function for the new tasks. Yu et al. (2022) and Hu et al. (2023) utilize reward-free data for improving offline learning but assume a labeled offline dataset is available for learning a reward function. Ye et al. (2022); Ghosh et al. (2023) consider the setting where reward and action are absent. We focus on the reward-free setting and leverage offline data to accelerate online learning.

**Unsupervised Behavior Extraction.** Many recent algorithms have been proposed for intrinsic behavioral learning without a reward. Popular methods includes prediction methods (Burda et al., 2018; Pathak et al., 2017, 2019), maximal entropy-based methods (Campos et al., 2021; Liu & Abbeel, 2021b,a; Mutti et al., 2020; Seo et al., 2021; Yarats et al., 2021b), and maximal mutual information-based methods (Eysenbach et al., 2018; Hansen et al., 2019; Liu & Abbeel, 2021a; Sharma et al., 2019). However, they require an online environment to achieve the diversity objective. Ajay et al. (2020); Singh et al. (2020) employ generative models to extract behaviors in the offline dataset but have a limited ability to enrich diversity.

**Offline-to-online learning and policy reuse.** Offline-to-online RL has been popular recently since it provides a promising paradigm to improve offline further learned policies. Nair et al. (2020) is among the first to propose a direct solution to offline-to-online RL. (Lee et al., 2022) proposes to use a balanced replay buffer and ensembled networks to smooth the offline-to-online transition; Zhang et al. (2023) proposes to use the expanded policy for offline-to-online RL. Ball et al. (2023) proposes to reuse the off-policy algorithm with modifications like ensembles and layer normalization. Zhang et al. (2022) proposes a critic-guided approach to reuse previously learned policies. Zhang et al. (2023) proposes a similar and simpler way to reuse previous policies.

**Imitation Learning** Our method is also related to imitation learning (Hussein et al., 2017). Popular imitation learning methods include behavior cloning methods Pomerleau (1988) and inverse RL methods Ho & Ermon (2016); Xiao et al. (2019); Dadashi et al. (2020). However, they assume the demonstration in the dataset to be near optimal, while our method works well with datasets of mediocre quality.

## 2 Preliminaries

### 2.1 Episodic Reinforcement Learning

We consider finite-horizon episodic Markov Decision Processes (MDPs), defined by the tuple $(\mathcal{S}, \mathcal{A}, H, \mathcal{P}, r)$, where $\mathcal{S}$ is a state space, $\mathcal{A}$ is an action space, $H$ is the horizon and $\mathcal{P} = \{\mathcal{P}_h\}_{h=1}^H$, $r = \{r_h\}_{h=1}^H$ are the transition function and reward function, respectively.

A policy $\pi = \{\pi_h\}_{h=1}^H$ specifies a decision-making strategy in which the agent chooses its actions based on the current state, i.e., $a_h \sim \pi_h(\cdot \,|\, s_h)$. The value function $V_h^\pi : \mathcal{S} \to \mathbb{R}$ is defined as the sum of future rewards starting at state $s$ and step $h \in [H]$, and similarly, the Q-value function, i.e.

$$V_h^\pi(s) = \mathbb{E}_\pi \Big[ \sum_{t=h}^H r_t(s_t, a_t) \,\Big|\, s_h = s \Big], \quad Q_h^\pi(s,a) = \mathbb{E}_\pi \Big[ \sum_{t=h}^H r_h(s_t, a_t) \,\Big|\, s_h = s, a_h = a \Big]. \quad (1)$$

where the expectation is with respect to the trajectory $\tau$ induced by policy $\pi$.

We define the Bellman operator as

$$(\mathbb{B}_h f)(s,a) = \mathbb{E}\big[ r_h(s,a) + f(s') \big], \quad (2)$$

for any $f : \mathcal{S} \to \mathbb{R}$ and $h \in [H]$. The optimal Q-function $Q^*$, optimal value function $V^*$ and optimal policy $\pi^*$ are related by the Bellman optimality equation

$$V_h^*(s) = \max_{a \in \mathcal{A}} Q_h^*(s,a), \quad Q_h^*(s,a) = (\mathbb{B}_h V_h^*)(s,a), \quad \pi_h^*(\cdot \,|\, s) = \operatorname*{argmax}_\pi \mathbb{E}_{a \sim \pi} Q_h^*(s,a). \quad (3)$$

We define the suboptimality, as the performance difference of the optimal policy $\pi^*$ and the current policy $\pi_k$ given the initial state $s_1 = s$. That is

$$\mathrm{SubOpt}(\pi; s) = V_1^{\pi^*}(s) - V_1^\pi(s).$$

### 2.2 Linear Function Approximation

To derive a concrete bound, we consider the *linear MDP* (Jin et al., 2020, 2021) as follows, where the transition kernel and expected reward function are linear with respect to a feature map.

**Definition 2.1** (Linear MDP). MDP$(\mathcal{S}, \mathcal{A}, \mathrm{H}, \mathbb{P}, \mathrm{r})$ is a *linear MDP* with a feature map $\phi : \mathcal{S} \times \mathcal{A} \to \mathbb{R}^d$, if for any $h \in [H]$, there exist $d$ *unknown* (signed) measures $\mu_h = (\mu_h^{(1)}, \ldots, \mu_h^{(d)})$ over $\mathcal{S}$ and an *unknown* vector $z_h \in \mathbb{R}^d$, such that for any $(s,a) \in \mathcal{S} \times \mathcal{A}$, we have

$$\mathbb{P}_h(\cdot \,|\, s,a) = \langle \phi(s,a), \mu_h(\cdot) \rangle, \qquad r_h(s,a) = \langle \phi(s,a), z_h \rangle. \quad (4)$$

Without loss of generality, we assume $\|\phi(s,a)\| \leq 1$ for all $(s,a) \in \mathcal{S} \times \mathcal{A}$, and $\max\{\|\mu_h(\mathcal{S})\|, \|z_h\|\} \leq \sqrt{d}$ for all $h \in [H]$.

When emphasizing the dependence over the reward parameter $z$, we also use $r_h(s,a,z), V_h^\pi(s,z)$ and $V_h^*(s,z)$ to denote the reward function, the policy value and the optimal value under parameter $z$, respectively.

## 3 Method

*How can we effectively leverage abundant offline reward-free data for improved performance in online tasks?* Despite the absence of explicit supervision signals in such datasets, they may contain valuable behaviors with multiple modes. These behaviors can originate from human expertise or the successful execution of previous tasks. To extract these behaviors, we assume that each behavior corresponds to specific intentions, which are instantiated by corresponding reward functions. Then, by employing a standard offline RL algorithm, we can extract desired behavior modes from the offline dataset using their associated reward functions. Such behavior can be reused for online fine-tuning with methods such as policy expansion (Zhang et al., 2023) or policy reuse (Zhang et al., 2022).

Then, the question becomes how to specify possible intentions for the dataset. Surprisingly, we found that employing a random prior over the intentions can extract diverse and useful behaviors effectively in practice, which form the offline part of our method. The overall pipeline is illustrated in Figure 1. Initially, we utilize a random intention prior to generating diverse and random rewards, facilitating the learning of various modes of behavior. Subsequently, standard offline algorithms are applied to extract behaviors with generated reward functions. Once a diverse set of behaviors is collected, we integrate them with policy reuse techniques to perform online fine-tuning.

Formally, we define $\mathcal{Z}$ as the space of intentions, where each intention $z \in \mathcal{Z}$ induces a reward function $r_z : \mathcal{S} \times \mathcal{A} \to \mathbb{R}$ that defines the objective for an agent. Let the prior over intentions be $\beta \in \Delta(\mathcal{Z})$. In our case, we let $\mathcal{Z}$ be the weight space $\mathcal{W}$ of the neural network and $\beta$ be a distribution over the weight $w \in \mathcal{W}$ of the neural network $f_w$ and sample $N$ reward functions from $\beta$ independently. Specifically, we have

$$r_i(s, a) = f_{w_i}(s, a), \ w_i \sim \beta, \ \forall i = 1, \ldots, N.$$

We choose $\beta$ to be a standard initialization distribution for neural networks (Glorot & Bengio, 2010; He et al., 2015). We choose TD3+BC (Fujimoto & Gu, 2021) as our backbone offline algorithm, but our framework is general and compatible with any existing offline RL algorithms. Then, the loss function for the offline phase can be written as

$$\mathcal{L}_{\text{critic}}^{\text{offline}}(\theta_i) = \mathbb{E}_{(s,a,r_i,s') \sim \mathcal{D}_i^{\text{off}}} \left[ \left( r_i + \gamma Q_{\theta_i'}(s', \widetilde{a}) - Q_{\theta_i}(s, a) \right)^2 \right], \tag{5}$$

$$\mathcal{L}_{\text{actor}}^{\text{offline}}(\phi_i) = -\mathbb{E}_{(s,a) \sim \mathcal{D}_i^{\text{off}}} \left[ \lambda Q_{\theta_i}(s, \pi_{\phi_i}(s)) - (\pi_{\phi_i}(s) - a)^2 \right], \tag{6}$$

where $\widetilde{a} = \pi_{\phi_i'}(s') + \epsilon$ is the smoothed target policy and $\lambda$ is the weight for behavior regularization. We adopt a discount factor $\gamma$ even for the finite horizon problem as commonly used in practice (Fujimoto et al., 2018).

As for the online phase, we adopt the policy expansion framework (Zhang et al., 2023) to reuse policies. We first construct an expanded policy set $\widetilde{\pi} = [\pi_{\phi_1}, \ldots, \pi_{\phi_N}, \pi_w]$, where $\{\phi_i\}_{i=1}^N$ are the weights of extracted behaviors from offline phase and $\pi_w$ is a new randomly initialized policy. We then use the critic as the soft policy selector, which generates a distribution over candidate policies as follows:

$$P_{\widetilde{\pi}}[i] = \frac{\exp \left( \alpha \cdot Q(s, \widetilde{\pi}_i(s)) \right)}{\sum_j \exp \left( \alpha \cdot Q(s, \widetilde{\pi}_j(s)) \right)}, \forall i \in \{1, \cdots, N+1\}, \tag{7}$$

where $\alpha$ is the temperature. We sample a policy $i \in \{1, \cdots, N+1\}$ according to $P_{\widetilde{\pi}}$ at each time step and follows $\pi_i$ to collect online data. The policy $\pi_w$ and $Q_\theta$-value network undergo continuous training using the online RL loss, such as TD3 (Fujimoto et al., 2018)):

$$\mathcal{L}_{\text{critic}}^{\text{online}}(\theta) = \mathbb{E}_{(s,a,r,s') \sim \mathcal{D}^{\text{on}}} \left[ (r + \gamma Q_{\theta'}(s', \widetilde{a}) - Q_\theta(s, a))^2 \right], \tag{8}$$

$$\mathcal{L}_{\text{actor}}^{\text{online}}(w) = -\mathbb{E}_{(s,a) \sim \mathcal{D}^{\text{on}}} \left[ Q_\theta(s, \pi_w(s)) \right]. \tag{9}$$

The overall algorithm is summarized in Algorithm 1 and 2. We highlight elements important to our approach in purple.

---

**Algorithm 1** Phase 1: Offline Behavior Extraction

---

1: **Require**: Behavior size $N$, offline reward-free dataset $\mathcal{D}^{\text{off}}$, prior intention distribution $\beta$
2: Initialize parameters of $N$ independent offline agents $\{Q_{\theta_i}, \pi_{\phi_i}\}_{i=1}^N$
3: **for** $i = 1, \cdots, N$ **do**
4:     Sample a reward function from priors $z_i \sim \beta$
5:     Reannotate $\mathcal{D}^{\text{off}}$ as $\mathcal{D}_i^{\text{off}}$ with reward $r_{z_i}$
6:     **for** each training iteration **do**
7:         Sample a random minibatch $\{\tau_j\}_{j=1}^B \sim \mathcal{D}_i^{\text{off}}$
8:         Calculate $\mathcal{L}_{\text{critic}}^{\text{offline}}(\theta_i)$ as in Equation (5) and update $\theta_i$
9:         Calculate $\mathcal{L}_{\text{actor}}^{\text{offline}}(\phi_i)$ as in Equation (6) and update $\phi_i$
10:    **end for**
11: **end for**
12: **Return** $\{\pi_{\phi_i}\}_{i=1}^N$

---

**Algorithm 2** Phase 2: Online Policy Reuse

---

1: **Require**: $\{\pi_{\phi_i}\}_{i=1}^N$, offline dataset $\mathcal{D}^{\text{off}}$, update-to-data ratio $G$
2: Initialize online agents $Q_\theta, \pi_w$ and replay buffer $\mathcal{D}^{\text{on}}$
3: Construct expanded policy set $\widetilde{\pi} = [\pi_{\phi_1}, \ldots, \pi_{\phi_N}, \pi_w]$
4: **for** each episode **do**
5:     Obtain initial state $s_1$ from the environment
6:     **for** step $t = 1, \cdots, T$ **do**
7:         Construct $P_{\widetilde{\pi}}$ According to Equation (7)
8:         Pick an policy to act $\pi_t \sim P_{\widetilde{\pi}}, a_t \sim \pi_t(\cdot|s_t)$
9:         Store transition $(s_t, a_t, r_t, s_{t+1})$ in $\mathcal{D}^{\text{on}}$
10:        **for** $g = 1, \cdots, G$ **do**
11:           Calculate $\mathcal{L}_{\text{critic}}^{\text{online}}(\theta)$ as in Equation (8) and update $\theta$
12:        **end for**
13:        Calculate $\mathcal{L}_{\text{actor}}^{\text{online}}(w)$ as in Equation (9) and update $w$
14:     **end for**
15: **end for**

---

## 4 Theoretical Analysis

*How sound is the random intent approach for behavior extraction?* This section aims to establish a theoretical characterization of our proposed method. We first consider the completeness of our method. We show that any behavior can be formulated as the optimal behavior under some given intent, and any behavior set that is covered by the offline dataset can be learned effectively given the corresponding intent set. Then we consider the coverage of random rewards. We show that with a reasonable number of random reward functions, the true reward function can be approximately represented by the linear combination of the random reward functions with a high probability. Hence our method is robust to the randomness in the intent sampling process.

### 4.1 Completeness of the Intent Method for Behavior Extraction

We first investigate the completeness of intent-based methods for behavior extraction. Formally, we have the following proposition.

**Proposition 4.1.** *For any behavior $\pi$, there exists an intent $z$ with reward function $r(\cdot, \cdot, z)$ such that $\pi$ is the optimal policy under $r(\cdot, \cdot, z)$. That is, for any $\pi = \{\pi_h\}_{h=1}^H, \pi_h : \mathcal{S} \to \Delta(\mathcal{A})$, there exists $z \in \mathcal{Z}$ such that*

$$V_h^\pi(s, z) = V_h^*(s, z), \ \forall (s, a, h) \in \mathcal{S} \times \mathcal{A} \times [H].$$

*Proof.* Please refer to Appendix A.1 for detailed proof. □

Proposition 4.1 indicates that any behavior can be explained as achieving some intent. This is intuitive since we can relabel the trajectory generated by any policy as successfully reaching the final state in

hindsight (Andrychowicz et al., 2017). With Proposition 4.1, we can assign an intent $z$ to any policy $\pi$ as $\pi_z$. However, it is unclear whether we can learn effectively from offline datasets given an intent set $\mathcal{Z}$. This is guaranteed by the following theorem.

**Theorem 4.2.** *Consider linear MDP as defined in Definition 2.1. With an offline dataset $\mathcal{D}$ with size $N$, and the PEVI algorithm (Jin et al., 2021), the suboptimality of learning from an intent $z \in \mathcal{Z}$ satisfies*

$$\text{SubOpt}(\widehat{\pi}; s, z) = \mathcal{O}\left(\sqrt{\frac{C_z^\dagger d^2 H^3 \iota}{N}}\right), \tag{10}$$

*for sufficiently large $N$, where $\iota = \log(dN|\mathcal{Z}|/\delta)$, $c$ is an absolute constant and*

$$C_z^\dagger = \max_{h \in [H]} \sup_{\|x\|=1} \frac{x^\top \Sigma_{\pi_z,h} x}{x^\top \Sigma_{\rho_h} x},$$

*with $\Sigma_{\pi_z,h} = \mathbb{E}_{(s,a) \sim d_{\pi_z,h}(s,a)}[\phi(s,a)\phi(s,a)^\top]$, $\Sigma_{\rho_h} = \mathbb{E}_{\rho_h}[\phi(s,a)\phi(s,a)^\top]$.*

*Proof.* Please refer to Appendix A.2 for detailed proof. □

$C_z^\dagger$ represents the maximum ratio between the density of empirical state-action distribution $\rho$ and the density $d_{\pi_z}$ induced from the policy $\pi_z$. Theorem 4.2 states that, for any behavior, $\pi_z$ that is well-covered by the offline dataset $\mathcal{D}$, we can use standard offline RL algorithm to extract it effectively, in the sense that learned behavior $\widehat{\pi}$ has a small suboptimality under reward function $r(\cdot, \cdot, z)$. Compared to standard offline result (Jin et al., 2021), Theorem 4.2 is only worse off by a factor of $\log |\mathcal{Z}|$, which enables us to learn multiple behaviors from a single dataset effectively, as long as the desired behavior mode is well contained in the dataset.

## 4.2 Coverage of Random Rewards

This section investigates the coverage property of random rewards. Intuitively, using random rewards may suffer from high variance, and it is possible that all sampled intents lead to undesired behaviors. Theorem 4.3 shows that such cases rarely happen since with a reasonable number of random reward functions, the linear combination of them can well cover the true reward function with a high probability.

**Theorem 4.3.** *Assume the reward function $r(s,a)$ admits a RKHS representation $\psi(s,a)$ with $\|\psi(s,a)\|_\infty \leq \kappa$ almost surely. Then with $N = c_0\sqrt{M}\log(18\sqrt{M}\kappa^2/\delta)$ random reward functions $\{r_i\}_{i=1}^N$, the linear combination of the set of random reward functions $\widehat{r}(s,a)$ can approximate the true reward function with error*

$$\mathbb{E}_{(s,a) \sim \rho}[\widehat{r}(s,a) - r(s,a)]^2 \leq c_1 \log^2(18/\delta)/\sqrt{M},$$

*with probability $1 - \delta$, where $M$ is the size of the offline dataset $\mathcal{D}$, $c_0$ and $c_1$ are universal constants and $\rho$ is the distribution that generates the offline dataset $\mathcal{D}$.*

*Proof.* Please refer to Appendix A.3 for detailed proof. □

Theorem 4.3 indicates that RL algorithms are insensitive to the randomness in the reward function used for learning, as long as we are using a reasonable number of such functions. This phenomenon is more significant in offline algorithms since they are conservative in nature to avoid over-generalization. This partly explains why using random rewards is sufficient for behavior extraction, and can even generate policies comparable to oracle rewards. Such a phenomenon is also observed in prior works (Shin et al., 2023; Li et al., 2023).

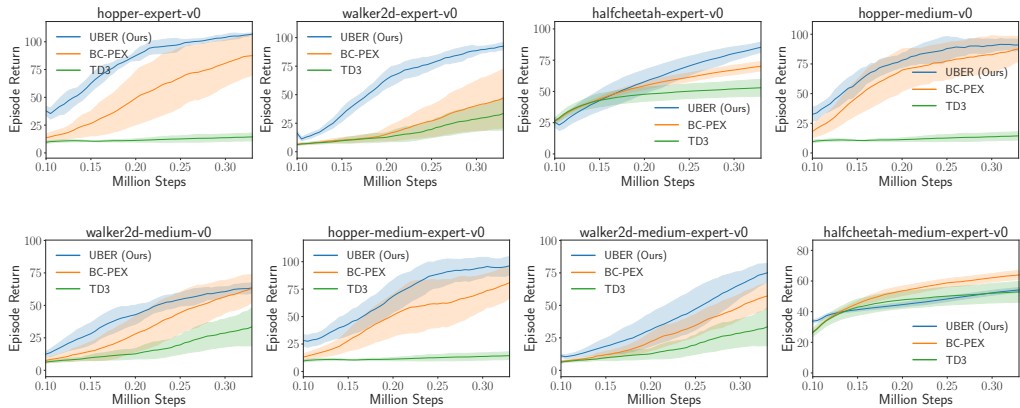

Figure 2: Comparison between UBER and baselines in the online phase in the Mujoco domain. We adopt datasets of various quality for offline training. We adopt a normalized score metric averaged with five random seeds.

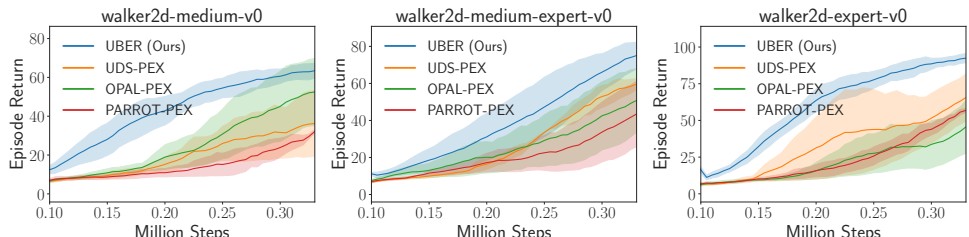

Figure 3: Comparison between UBER and baselines including unsupervised behavior extraction (PARROR OPAL) and data-sharing methods (UDS). The result is averaged with five random seeds. Our method outperforms these baselines by leveraging the diverse behavior set.

# 5 Experiments

Our evaluation studies UBER as a pre-training mechanism on reward-free data, focusing on the following questions: (1) Can we learn useful and diverse behaviors with UBER? (2) Can the learned behaviors from one task using UBER be reused for various downstream tasks? (3) How effective is each component in UBER?

To answer the questions above, we conduct experiments on the standard D4RL benchmark (Fu et al., 2020) and the multi-task benchmark Meta-World (Yu et al., 2020), which encompasses a variety of dataset settings and tasks. We adopt the normalized score metric proposed by the D4RL benchmark, and all experiments are averaged over five random seeds. Please refer to Appendix D for more experimental details.

## 5.1 Experiment Setting

We extract offline behavior policies from multi-domains in D4RL benchmarks, including locomotion and navigation tasks. Specifically, The locomotion tasks feature online data with varying levels of expertise. The navigation task requires composing parts of sub-optimal trajectories to form more optimal policies for reaching goals on a MuJoco Ant robot.

**Baselines.** We compare UBER with baselines with various behavior extraction methods. We compare our method with BC-PEX, which uses a behavior-cloning objective and other unsupervised behavior extraction methods (Yang et al., 2022), including OPAL (Ajay et al., 2020) and PARROT (Singh et al., 2020). We also compare our method with a strong offline-to-online method, RLPD (Ball et al., 2023) and an unsupervised data sharing method, UDS (Yu et al., 2022). Specifi-

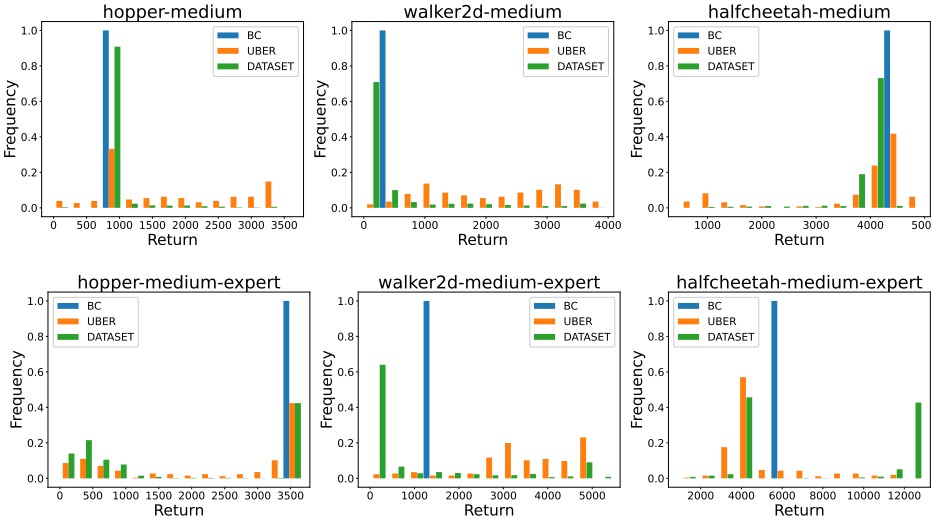

Figure 4: The return distribution of the dataset (DATASET) and different behaviors generated from random intent priors (UBER) and behavior cloning (BC). UBER can generate a diverse behavior set that span a return distribution wider than the original dataset.

| 500k step scores | RLPD (w\o true reward) | BC-PEX | UBER (Ours) | RLPD (w true reward) |
|---|---|---|---|---|
| umaze | 0.0±0.0 | **99.8±0.3** | 97.9±0.3 | 99.9±0.1 |
| umaze-diverse | 0.0±0.0 | **99.9±0.1** | 98.8±0.5 | 99.9±0.1 |
| medium-play | 0.0±0.0 | 0.0±0.0 | **94.0±3.1** | 98.7±0.9 |
| medium-diverse | 0.0±0.0 | 0.0±0.0 | **96.5±1.5** | 98.5±1.3 |
| large-play | 0.0±0.0 | 0.0±0.0 | **83.3±7.7** | 94.8±1.5 |
| large-diverse | 0.0±0.0 | 0.0±0.0 | **88.7±1.5** | 93.5±1.6 |
| *100k step scores* | RLPD (w\o true reward) | BC-PEX | UBER (Ours) | RLPD (w true reward) |
| umaze | 0.0±0.0 | **95.8±1.4** | 90.7±1.8 | 82.5±0.2 |
| umaze-diverse | 0.0±0.0 | **95.5±3.7** | 94.5±1.1 | 83.3±0.3 |
| medium-play | 0.0±0.0 | 0.0±0.0 | **91.9±2.9** | 79.5±0.6 |
| medium-diverse | 0.0±0.0 | 0.0±0.0 | **91.3±1.4** | 75.3±1.3 |
| large-play | 0.0±0.0 | 0.0±0.0 | **62.8±2.6** | 62.1±3.7 |
| large-diverse | 0.0±0.0 | 0.0±0.0 | **82.7±2.8** | 61.4±4.4 |

Table 1: Comparison between UBER and baselines in the online phase in the Antmaze domain at 100k and 500k environment steps. Pure online methods without an offline dataset fail completely due to the difficulty of exploration.

cally, BC-PEX first extracts behavior from the offline dataset based on behavior cloning and then conducts online policy reuse with PEX (Zhang et al., 2023) following the schedule of Algorithm 2. As for OPAL-PEX and PARROT-PEX, we extract behavior based on the VAE and FLOW models, and then conduct the same online policy reuse pipeline. We set the reward of the offline dataset to zero to implement UDS-PEX. As for the pure online baseline, we use RLPD without prior offline data, named RLPD (no prior) as our baseline.

## 5.2 Experimental Results

**Answer of Question 1:** To show that UBER can generate diverse behaviors, we plot the return distribution of behaviors generated by UBER, as well as the original return distribution from the dataset. The experimental results are summarized in Figure 4. We also provide the entropy of the return distribution in Table 3 for a numerical comparison. We can see that (1) UBER can extract policies that perform better than the dataset, especially when the dataset does not have high-quality

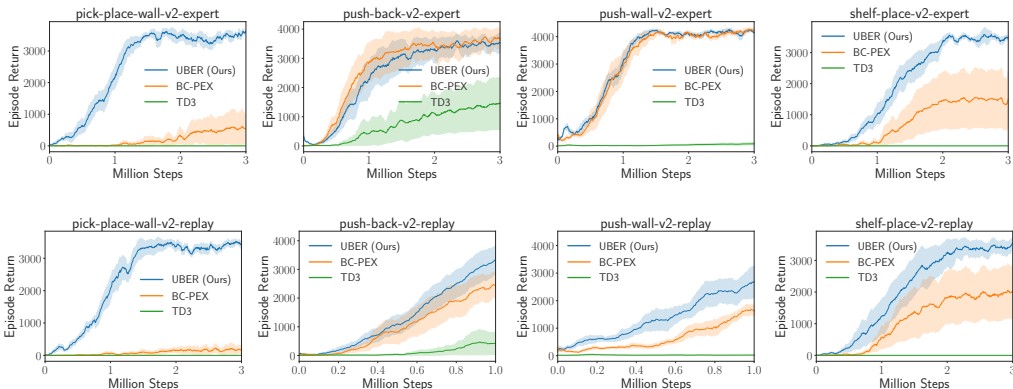

Figure 5: Experimental results on the meta-world based on two types of offline datasets, expert and replay. All experiment results were averaged over five random seeds. Our method achieves better or comparable results than the baselines consistently.

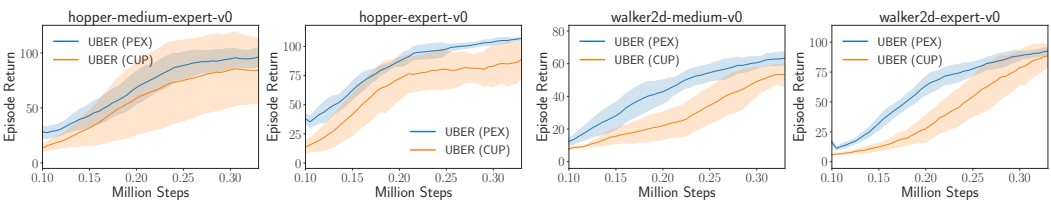

Figure 6: Ablation study for the online policy reuse module. Using policy expansion (PEX) for downstream tasks is better than critic-guided policy reuse (CUP) in general.

data; (2) UBER can learn a diverse set of behaviors, spanning a wider distribution than the original dataset. We hypothesize that diversity is one of the keys that UBER can outperform behavior cloning-based methods.

Further, we conduct experiments to test if the set of random intents can cover the true intent. We calculate the correlation of $N = 256$ random rewards with the true reward and measure the linear projection error. The results in Table 2 indicate that random intents can have a high correlation with true intent, and linear combinations of random rewards can approximate the true reward function quite well.

To test whether UBER can learn useful behaviors for downstream tasks, we compare UBER with BC-PEX, AVG-PEX, and TD3 in the online phase in the Mujoco domains. The experimental results in Figure 2 show that our method achieves superior performance than baselines. By leveraging the behavior library generated from UBER, the agent can learn much faster than learning from scratch, as well as simple behavior cloning methods. Further, the results in Figure 3 show that UBER also performs better than behavior extraction and data sharing baselines in most tasks. This is because prior methods extract behaviors in a behavior-cloning manner, which lacks diversity and leads to degraded performance for downstream tasks.

In addition, to test UBER's generality across various offline and online methods on various domains, we also conduct experiments on Antmaze tasks. We use IQL (Kostrikov et al., 2021) and RLPD (Ball et al., 2023) as the backbone of offline and online algorithms. The experimental results in Table 1 show that our method achieves stronger performance than baselines. RLPD relies heavily on offline data and RLPD without true reward has zero return on all tasks. Differently, UBER extracts a useful behavior policy set for the online phase, achieving similar sample efficiency as the oracle method that has reward label for the offline dataset. We also provide the learning curve for the antmaze environment in Figure 11.

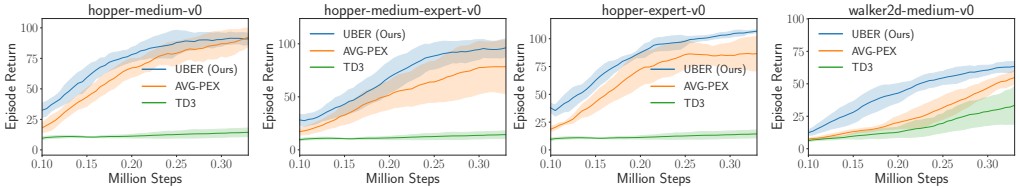

Figure 7: Ablation study for the random intent priors module. Using average reward for policies learning can lead to nearly optimal performances on the original task. Nevertheless, our method outperforms such a baseline (AVG-PEX) consistently, showing the importance of behavioral diversity.

**Answer of Question 2:** To test whether the behavior set learned from UBER can benefit multiple downstream tasks, we conducted the multi-task experiment on Meta-World (Yu et al., 2020), which requires learning various robot skills to perform various manipulation tasks. We first extract behaviors with random intent priors from the selected tasks and then use the learned behaviors for various downstream tasks. The experimental results in Appendix B show that the prior behaviors of UBER can be successfully transferred across multi-tasks than baselines. Please refer to Appendix B for the experimental details and results.

**Answer of Question 3:** To understand what contributes to the performance of UBER, we perform ablation studies for each component of UBER. We first replace the policy expansion module with critic-guided policy reuse (CUP; Zhang et al., 2022), to investigate the effect of various online policy reuse modules. The experimental results in Figure 6 show that UBER+PEX generally outperforms UBER+CUP. We hypothesize that CUP includes a hard reuse method, which requires the optimized policy to be close to the reused policies during training, while PEX is a soft method that only uses the pre-trained behaviors to collect data. This makes PEX more suitable to our setting since UBER may generate undesired behaviors.

Then, we ablate the random intent priors module. We use the average reward in the offline datasets to learn offline policies and then follow the same online policy reuse process in Algorithm 2. Using average reward serves as a strong baseline since it may generate near-optimal performance as observed in Shin et al. (2023). We name the average reward-based offline behavior extraction method as AVG-PEX. The experimental results in Figure 7 show that while average reward-based offline optimization can extract some useful behaviors and accelerate the downstream tasks, using random intent priors performs better due to the diversity of the behavior set.

## 6 Conclusion

This paper presents Unsupervised Behavior Extraction via Random Intent Priors (UBER), a novel approach to enhance reinforcement learning using unsupervised data. UBER leverages random intent priors to extract diverse and beneficial behaviors from offline, reward-free datasets. Our theorem justifies the seemingly simple approach, and our experiments validate UBER's effectiveness in generating high-quality behavior libraries, outperforming existing baselines, and improving sample efficiency for deep reinforcement learning algorithms.

UBER unlocks the usage of abundant reward-free datasets, paving the way for more practical applications of RL. UBER focuses on learning behaviors, which is orthogonal to representation learning. It is an interesting future direction to further boost online sample efficiency by combining both approaches. Consuming large unsupervised datasets is one of the keys to developing generalist and powerful agents. We hope the principles and techniques encapsulated in UBER can inspire further research and development in unsupervised reinforcement learning.

## 7 Acknowledegement

This work is supported in part by Science and Technology Innovation 2030 - "New Generation Artificial Intelligence" Major Project (No. 2018AAA0100904) and the National Natural Science Foundation of China (62176135).

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

# A  Missing Proofs

## A.1  Proof of Proposition 4.1

**Proposition A.1** (Proposition 4.1 restated). *For any behavior $\pi$, there exists an intent $z$ with reward function $r(\cdot, \cdot, z)$ such that $\pi$ is the optimal policy under $r(\cdot, \cdot, z)$. That is, for any $\pi = \{\pi_h\}_{h=1}^H, \pi_h : \mathcal{S} \to \Delta(\mathcal{A})$, there exists $z \in \mathcal{Z}$ such that*

$$V_h^\pi(s, z) = V_h^*(s, z), \ \forall (s, a, h) \in \mathcal{S} \times \mathcal{A} \times [H].$$

*Moreover, if $\pi$ is deterministic, then there exists $z \in \mathcal{Z}$ such that $\pi$ is the unique optimal policy under $r(\cdot, \cdot, z)$, in the sense that for all optimal policy $\pi_z^*, \pi_{h,z}^*(\cdot|s) = \pi_h(\cdot|s), \forall d_h^\pi(s) > 0$.*

*Proof.* For any policy $\pi$ and any reward function $r_h(\cdot, \cdot, z)$, we have

$$V_h^\pi(s_h, z) = \sum_{t=h}^H \int r_t(s, a, z) d_t^\pi(s, a, s_h) \mathrm{d}s \mathrm{d}a, \tag{11}$$

where $d_t^\pi(s, a, s_h)$ is the visitation density at step $t$ for $\pi$ starting from $s_h$.

Let $r_h(s, a, z) = \mathbb{1}(\exists s_h \in \mathcal{S}, d_h^\pi(s, a, s_h) > 0)$, then we have

$$V_h^\pi(\cdot, z) = H - h = V_{h,\max} = V_h^*(\cdot, z). \tag{12}$$

If $\pi$ is deterministic, then $d_h^\pi(s, a)$ is a one-hot distribution at any given $s$. So any $\pi'$ such that there exists $s_h, \pi_h(\cdot|s_h) \neq \pi_h'(\cdot|s_h)$ and $d_h^\pi(s_h) > 0$, we have

$$\int r_h(s_h, a, z) d_h^{\pi'}(s_h, a) \mathrm{d}a < 1,$$

Then we have $V_h^{\pi'}(s_h, z) < V_{h,\max}$, which means $\pi'$ is not optimal. Therefore, the optimal policy $\pi$ is unique.

$\square$

Proposition 4.1 shows that any deterministic policy can be uniquely determined by some intent $z$ so that it is extractable via standard RL.

## A.2  Proof of Theorem 4.2

**Theorem A.2** (Theorem 4.2 restated.). *Consider linear MDP as defined in Definition 2.1. With an offline dataset $\mathcal{D}$ with size $N$, and the PEVI algorithm (Jin et al., 2021), the suboptimality of learning from an intent $z \in \mathcal{Z}$ with size $|\mathcal{Z}|$ satisfies*

$$\mathrm{SubOpt}(\widehat{\pi}; r_z) \leq 4c \sqrt{\frac{C_z^\dagger d^2 H^3 \iota}{N}}, \tag{13}$$

*with probability $1 - \delta$ for sufficiently large $N$, where $\iota = \log \frac{4dN|\mathcal{Z}|}{\delta}$ is a logarithmic factor, $c$ is an absolute constant and*

$$C_z^\dagger = \max_{h \in [H]} \sup_{\|x\|=1} \frac{x^\top \Sigma_{\pi_z, h} x}{x^\top \Sigma_{\rho_h} x},$$

*with*

$$\Sigma_{\pi_z, h} = \mathbb{E}_{(s,a) \sim d_{\pi_z, h}(s,a)}[\phi(s,a)\phi(s,a)^\top], \quad \Sigma_{\rho_h} = \mathbb{E}_{\rho_h}[\phi(s,a)\phi(s,a)^\top].$$

*Proof.* Using the fact that any linear reward function is still a linear MDP, we can reuse the proof for standard offline RL.

Let $\delta' = \frac{\delta}{|\mathcal{Z}|}$ and following Jin et al. (2021), we have the result immediately with a union bound.

Note that the original bound in Jin et al. (2021) scales as $\widetilde{\mathcal{O}}(d^3)$, but it can be improved to $\widetilde{\mathcal{O}}(d^2)$ by finer analysis without changing the algorithm (Xiong et al., 2022). $\square$

### A.3 Proof of Theorem 4.3

*Proof.* Each random reward function can be seen as one dimension of random feature for linear regression. Following Theorem 1 in Rudi & Rosasco (2017), and note that the true reward function has zero error, we have that the linear estimator $\widehat{r}_{\text{ridge}}$ generated by ridge regression satisfy

$$\mathbb{E}_{(s,a)\sim\rho}(\widehat{r}_{\text{ridge}}(s,a) - r(s,a))^2 = \mathbb{E}_{(s,a)\sim\rho}\left(\sum_i \widehat{w}_i^{\text{ridge}} r_i(s,a) - r(s,a)\right)^2 \leq c_1 \log^2(18/\delta)/\sqrt{M}.$$

Noting that

$$\min_w \mathbb{E}_{(s,a)\sim\rho}\left(\sum_i w_i r_i(s,a) - r(s,a)\right)^2 \leq \mathbb{E}_{(s,a)\sim\rho}\left(\sum_i \widehat{w}_i^{\text{ridge}} r_i(s,a) - r(s,a)\right)^2,$$

we have result immediately.

$\square$

## B Experiments on Multi-task Transfer

**Answer of Question 2:** We evaluate our method on Meta-World (Yu et al., 2020), a popular reinforcement learning benchmark composed of multiple robot manipulation tasks. These tasks are correlated (performed by the same Sawyer robot arm) while being distinct (interacting with different objectives and having different reward functions). Following Zhang et al. (2022), we use datasets from three representative tasks: Reach, Push, and Pick-Place, as shown in Figure 8 and we choose Push-Wall, Pick-Place-Wall, Push-Back, and Shelf-Place as the target tasks.

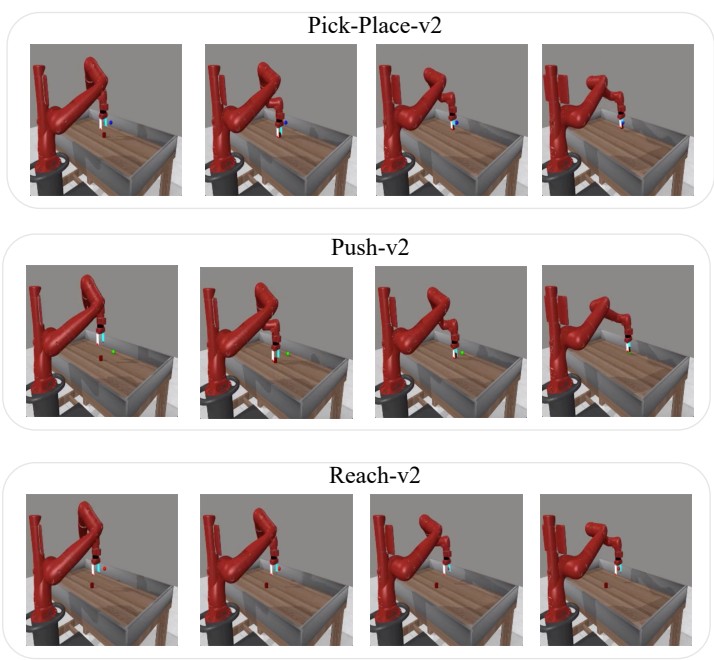

Figure 8: Visualization of three representative tasks in the metaworld.

Two types of offline datasets were considered: an expert policy dataset and a replay dataset generated by TD3. The expert dataset comprised 0.1M data points obtained from expert demonstrations, while the replay dataset consisted of 0.3M data points sampled from 3M TD3 experiences.

Our experimental results, depicted in Figure 9, reveal that UBER outperforms the baselines in most tasks. This underscores UBER's ability to effectively repurpose learned behaviors from various tasks for different downstream applications.

# C Additional Experiments

**Correlation and projection error of random rewards.** To investigate whether random rewards has a good coverage and a high correlation with the true reward, we calculate the maximum correlation between random rewards and the true reward, as well as the projection error of the true reward on the linear combination of random rewards on Mujoco tasks, as shown in Table 2.

| Tasks | Max Correlation | Min Correlation | Projection Error |
|---|---|---|---|
| hopper-medium-v0 | 0.569 | -0.568 | 0.016 |
| hopper-medium-expert-v0 | 0.498 | -0.540 | 0.015 |
| hopper-expert-v0 | 0.423 | -0.415 | 0.011 |
| halfcheetah-medium-v0 | 0.569 | -0.568 | 0.016 |
| halfcheetah-medium-expert-v0 | 0.605 | -0.589 | 0.047 |
| halfcheetah-expert-v0 | 0.370 | -0.461 | 0.021 |
| walker2d-medium-v0 | 0.475 | -0.582 | 0.046 |
| walker2d-medium-expert-v0 | 0.495 | -0.472 | 0.042 |
| walker2d-expert-v0 | 0.358 | -0.503 | 0.019 |

Table 2: Minimum and maximum correlation and linear projection error for the true reward function using random reward functions on various tasks. The projection error $\epsilon$ is defined as $\epsilon = \|r - \widehat{r}\|/\|r\|$, where $\widehat{r}$ is the best approximation using linear combination of random rewards. Note that we use random reward functions (represented by neural networks) based on $(s, a)$ inputs rather than entirely random ones, which have a near-zero correlation with the true reward and $40\%$ projection error.

**Entropy of return distribution.** To numerically show that UBER can generate a diverse behavior set beyond the original dataset, we calculate the entropy of the return distribution for each task, as shown in Table 3.

| Methods | halfcheetah-m | halfcheetah-me | halfcheetah-e | hopper-m | hopper-me | hopper-e |
|---|---|---|---|---|---|---|
| BC | 0 | 0 | 0 | 0 | 0 | 0 |
| DATASET | 1.21 | 1.47 | 0.86 | 0.71 | 1.97 | 0.82 |
| UBER | 2.01 | 1.92 | 2.69 | 2.45 | 2.32 | 1.11 |

| Methods | walker2d-m | walker2d-me | walker2d-e | antmaze-md | antmaze-mr |
|---|---|---|---|---|---|
| BC | 0 | 0 | 0 | 0 | 0 |
| DATASET | 1.44 | 1.88 | 2.43 | 0.63 | 0.29 |
| UBER | 2.67 | 2.61 | 0.41 | 1.77 | 1.69 |

Table 3: Entropy of the return distribution for different methods. Our method generate a behavior set that has a higher entropy in the return distribution than the original dataset consistently across various tasks.

**Distribution of behaviors.** To show that UBER can generate diverse behaviors, we plot the distribution of behaviors generated by UBER, as well as the original distribution from the dataset. The experimental results are summarized in Figure 10. We can see that when there are expert behaviors, most of the behaviors generated by UBER successfully lock to the optimal policy. When there are multiple modes of behavior like `medium-expert`, UBER can learn both behaviors, matching the distribution of the dataset. When there are no expert behaviors, UBER can learn diverse behaviors, some of which can achieve near-optimal performance. We hypothesize that diversity is one of the keys that UBER can outperform behavior cloning-based methods.

**Learning curve for the Antmaze environment.** Here we provide the detailed learning curve for the Antmaze experiment to demonstrate the efficiency of UBER. We can see from Figure 11 that UBER learns a good strategy in hard exploration tasks like `medium` and `large`, indicating UBER's strong ability to learn useful behaviors.

**Ablation study for the online policy reuse module.** We first replace the policy expansion module with CUP (Zhang et al., 2022), UBER (CUP) to investigate the effect of various online policy reuse modules. The experimental results in Figure 12 show that UBER+PEX generally outperforms UBER+CUP. We hypothesize that CUP includes a hard reuse method, which requires the optimized policy to be close to the reused policies during training, while PEX is a soft method that only uses the pre-trained behaviors to collect data. This makes PEX more suitable to our setting since UBER may generate undesired behaviors.

**Complete comparison with baselines.** Here we provide a complete comparison with several baselines, including OPAL, PARROT and UDS, as shown in Figure 13. Our method outperform these baselines due to the ability to extract diverse behaviors.

**Ablation study for the random intent priors module.** we conduct an ablation study for the random intent priors module. We use the average reward in the offline datasets to learn offline policies and then follow the same online policy reuse process in Algorithm 2. Using average reward serves as a strong baseline since it may generate near-optimal performance as observed in Shin et al. (2023). We name the average reward-based offline behavior extraction as AVG-PEX. The experimental results in Figure 14 show that while average reward-based offline optimization can extract some useful behaviors and accelerate the downstream tasks, using random intent priors performs better due to the diversity of the behavior set.

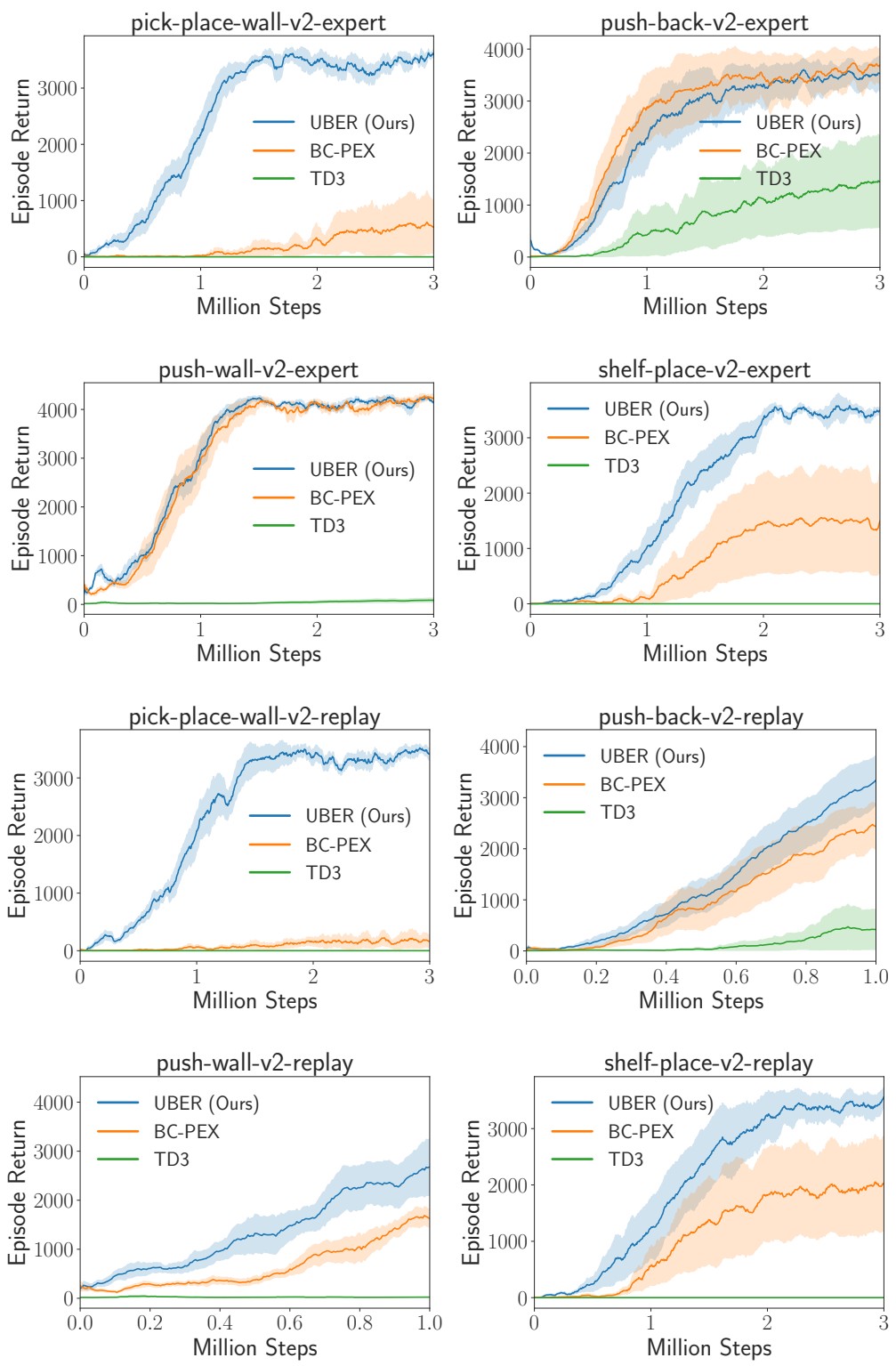

Figure 9: Complete experimental results on the meta-world based on two types of offline datasets, expert and replay. All experiment results were conducted over five random seeds.

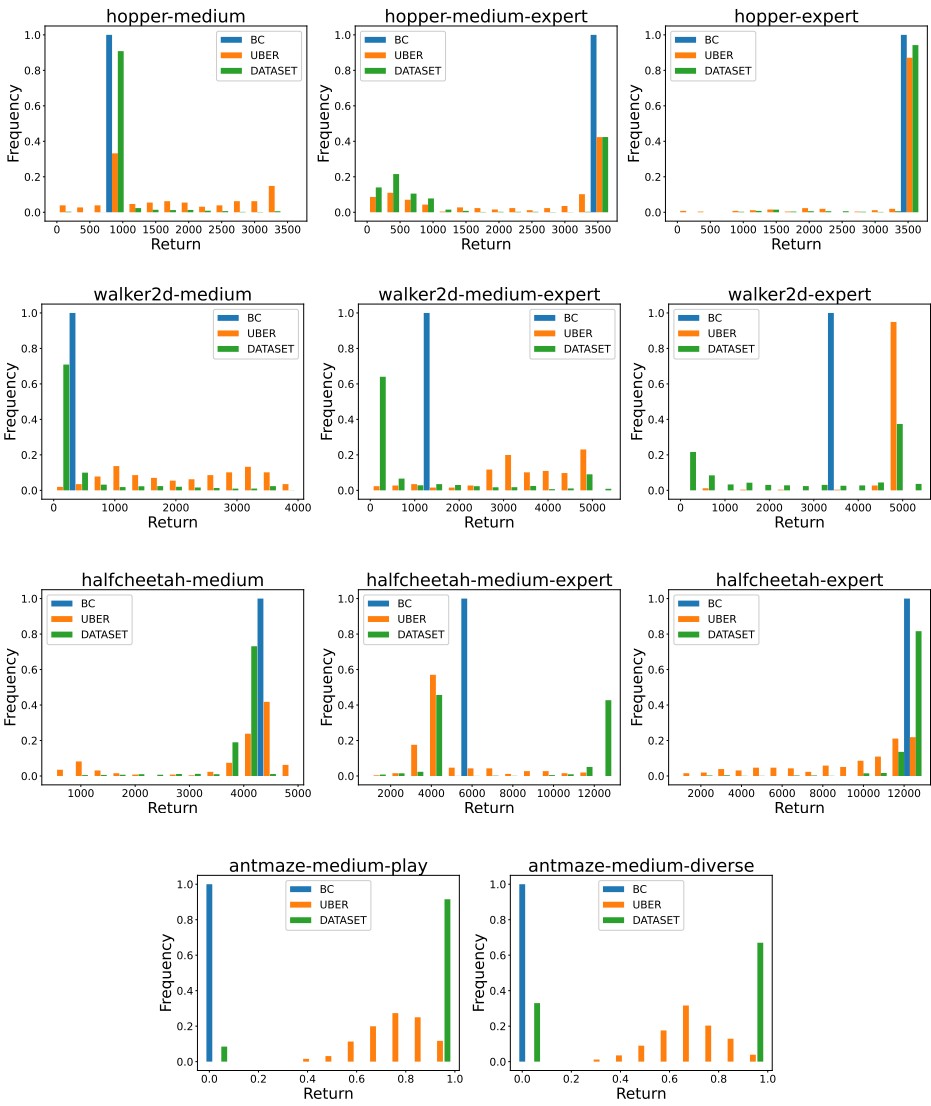

Figure 10: Distribution of random intent priors, datasets, and behavior cloning policies. We restrict the y-axis to a small range to make the distribution clearer.

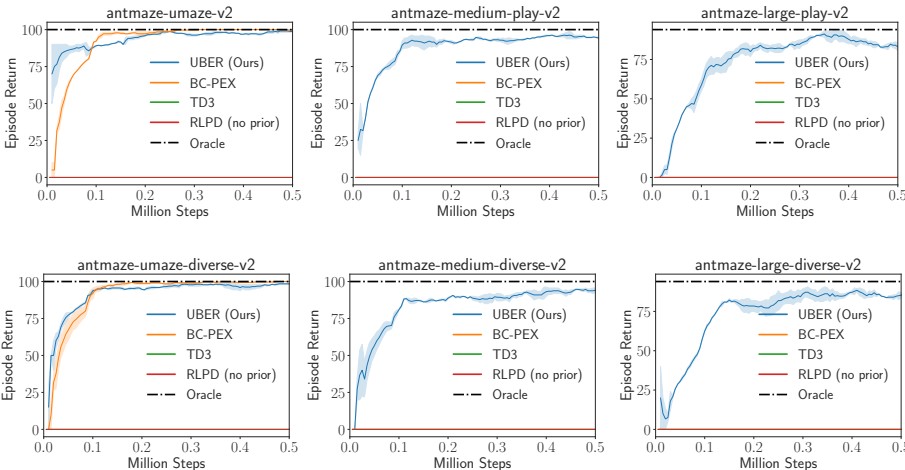

Figure 11: Comparison between UBER and baselines in the online phase in the Antmaze domain. Oracle is the performance of RLPD at 500k step. We adopt a normalized score metric averaged with three random seeds.

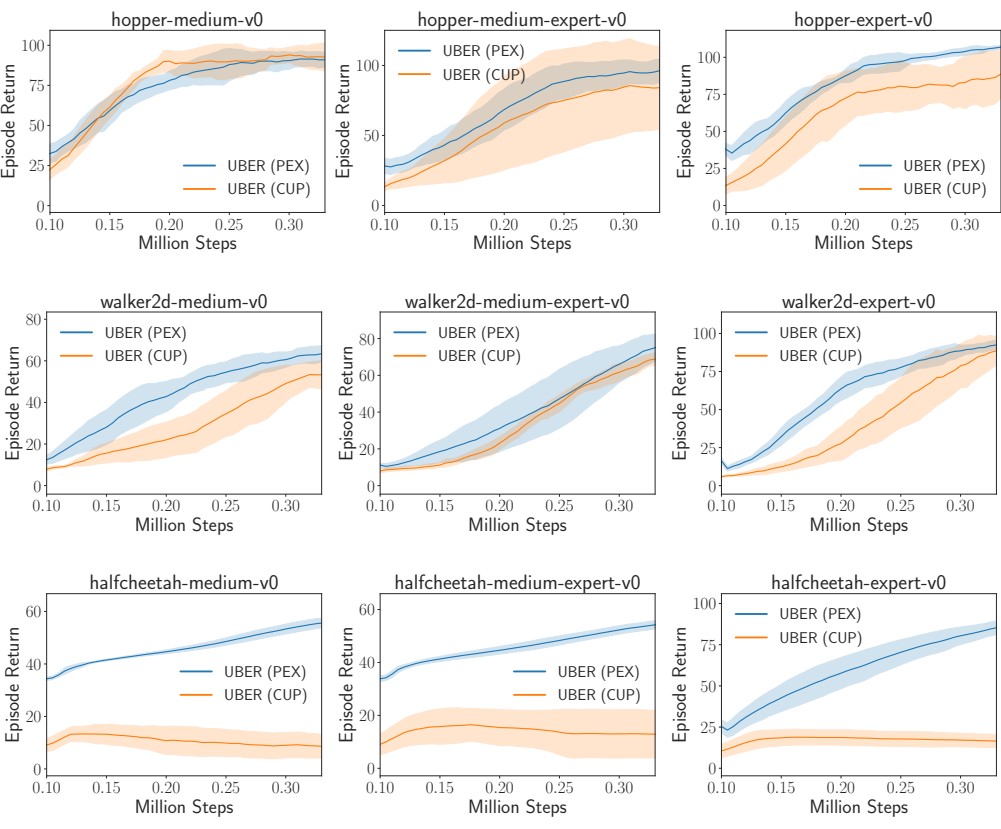

Figure 12: Complete result for the ablation study for the online policy reuse module. Using policy expansion (PEX) for downstream tasks is better than critic-guided policy reuse (CUP) in general

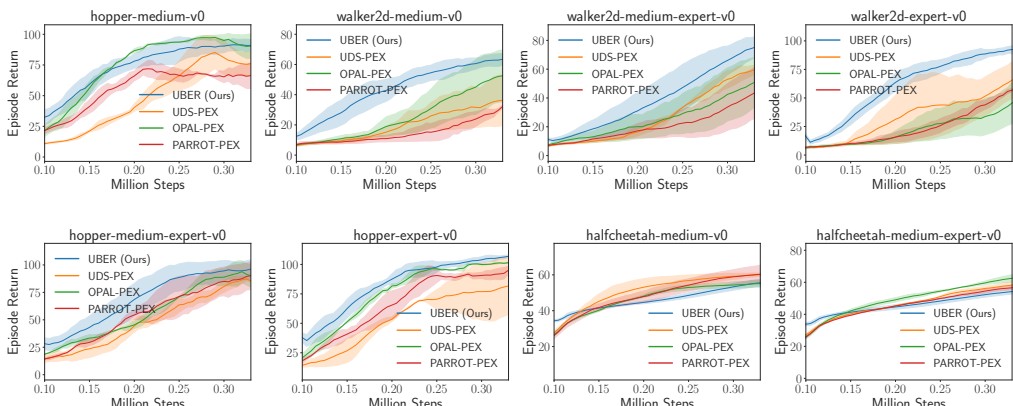

Figure 13: Complete result for the comparison between UBER and baselines including unsupervised behavior extraction and data-sharing method. We adopt a normalized score metric averaged with five random seeds.

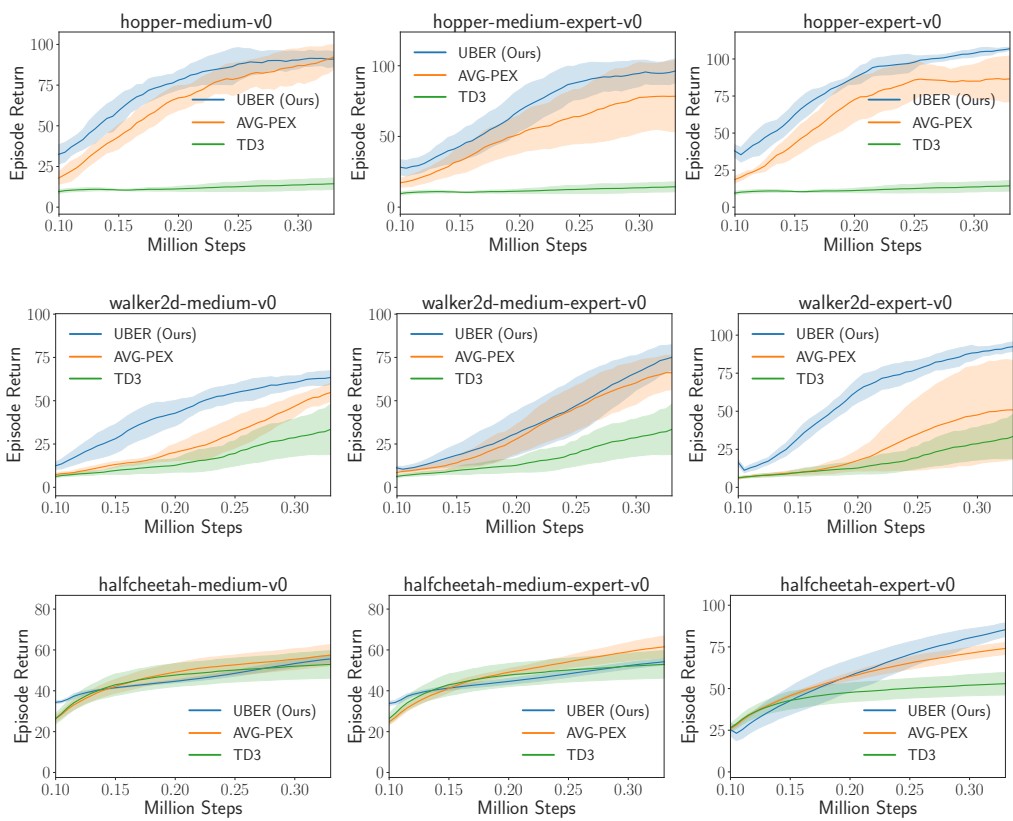

Figure 14: Complete result for the ablation study for the random intent priors module. Using average reward for policies learning can lead to nearly optimal performances on the original task. Nevertheless, our method outperforms such a baseline (AVG-PEX) consistently, showing the importance of behavioral diversity.

# D   Experimental Details

**Experimental Setting.**   For BC-PEX, we first extract behavior from the offline dataset based on the Behavior Cloning, then conduct online policy expansion based on Algorithm 2. For Avg-PEX, we first use the average reward for the offline optimization, then conduct online policy expansion based on Algorithm 2. For RLPD, we load the offline dataset with the true reward value for the prior data. For RLPD (no prior), we do not load offline-dataset into the buffer.

**Hyper-parameters.**   For the Mujoco and metaworld tasks, we adopt the TD3+BC and TD3 as the backbone of offline and online algorithms. For the Antmaze tasks, we adopt the IQL and RLPD as the backbone of offline and online algorithms. We outline the hyper-parameters used by UBER in Table 4, Table 5 and Table 6.

| Hyperparameter | Value |
|---|---|
| Optimizer | Adam |
| Critic learning rate | 3e-4 |
| Actor learning rate | 3e-4 |
| Mini-batch size | 256 |
| Discount factor | 0.99 |
| Target update rate | 5e-3 |
| Policy noise | 0.2 |
| Policy noise clipping | (-0.5, 0.5) |
| TD3+BC parameter $\alpha$ | 2.5 |
| **Architecture** | **Value** |
| Critic hidden dim | 256 |
| Critic hidden layers | 2 |
| Critic activation function | ReLU |
| Actor hidden dim | 256 |
| Actor hidden layers | 2 |
| Actor activation function | ReLU |
| Random Net hidden dim | 256 |
| Random Net hidden layers | 2 |
| Random Net activation function | ReLU |
| **UBER Parameters** | **Value** |
| Random reward dim $n$ | 256 |
| PEX temperature $\alpha$ | 10 |

Table 4: Hyper-parameters sheet of UBER in Mujoco tasks.

**Baselines Implementation.**   We adopt the author-provided implementations from GitHub for TD3 [*], TD3+BC [†], IQL [‡], RLPD [§] and DrQ-v2 [¶]. All experiments are conducted on the same experimental setup, a single GeForce RTX 3090 GPU and an Intel Core i7-6700k CPU at 4.00GHz.

---

[*] `https://github.com/sfujim/TD3`

[†] `https://github.com/sfujim/TD3_BC`

[‡] `https://github.com/ikostrikov/implicit_q_learning`

[§] `https://github.com/ikostrikov/rlpd`

[¶] `https://github.com/facebookresearch/drqv2`

| Hyperparameter | Value |
| --- | --- |
| Optimizer | Adam |
| Critic learning rate | 3e-4 |
| Actor learning rate | 3e-4 |
| Mini-batch size | 256 |
| Discount factor | 0.99 |
| Target update rate | 5e-3 |
| Policy noise | 0.2 |
| Policy noise clipping | (-0.5, 0.5) |
| TD3+BC parameter $\alpha$ | 2.5 |
| Architecture | Value |
| Critic hidden dim | 256 |
| Critic hidden layers | 2 |
| Critic activation function | ReLU |
| Actor hidden dim | 256 |
| Actor hidden layers | 2 |
| Actor activation function | ReLU |
| Random Net hidden dim | 256 |
| Random Net hidden layers | 2 |
| Random Net activation function | ReLU |
| UBER Parameters | Value |
| Random reward dim $n$ | 100 |
| PEX temperature $\alpha$ | 10 |

Table 5: Hyper-parameters sheet of UBER in metaworld tasks.

| Hyperparameter | Value |
| --- | --- |
| Optimizer | Adam |
| Critic learning rate | 3e-4 |
| Actor learning rate | 3e-4 |
| Mini-batch size | 256 |
| Discount factor | 0.99 |
| Target update rate | 5e-3 |
| IQL parameter $\tau$ | 0.9 |
| RLPD parameter $G$ | 20 |
| Ensemble Size | 10 |
| Architecture | Value |
| Critic hidden dim | 256 |
| Critic hidden layers | 2 |
| Critic activation function | ReLU |
| Actor hidden dim | 256 |
| Actor hidden layers | 2 |
| Actor activation function | ReLU |
| Random Net hidden dim | 256 |
| Random Net hidden layers | 2 |
| Random Net activation function | ReLU |
| UBER Parameters | Value |
| Random reward dim $n$ | 256 |
| PEX temperature $\alpha$ | 10 |

Table 6: Hyper-parameters sheet of UBER in Antmaze tasks.

