# OpenReview forum: "Unsupervised Behavior Extraction via Random Intent Priors"
_NeurIPS.cc/2023/Conference — NeurIPS 2023 poster_

### Official Review · Reviewer_XkfL · 2023-07-07

**Soundness:** 3 good
**Presentation:** 2 fair
**Contribution:** 3 good
**Rating:** 6
**Confidence:** 3

**Summary:**

The authors propose UBER, a method for learning a collection of behavior policies from offline experience data lacking reward labels and ultimately adapting these behaviors in an online setting. UBER generates a collection of randomly-initialized reward models and trains a policy on the offline data using each. During online adaptation, UBER jointly learns an action-value function and policy, selecting between the pre-trained policies to collect data using the critic. In several simulated offline-to-online settings, UBER performs similarly or better than some existing algorithms.

**Strengths:**

The paper studies an important and relevant problem, offline-to-online RL, and describes a simple but interesting approach to pre-training useful behavior policies for online exploration. The results are fairly promising, showing that UBER outperforms existing approaches to online adaptation after pre-training on online data. The experiments cover a decent collection of environments.

**Weaknesses:**

While I appreciate an attempt to rigorously understand the proposed method, the interpretations in the theoretical sections seem a bit too generous to me. For example, the suboptimality bound in Eq. 8 seems very loose (proportional to $\sqrt d^3H^3$), requiring a massive number of samples for fairly modestly state spaces and horizons. Further, the condition in Proposition 4.3 that the infinity norm of the difference of the reward functions is small actually seems very strong to me, and it seems fairly intuitive that the values of the optimal policies under each wouldn't change much. There is no attempt to show experimentally that this norm is small for the actual random reward functions used in practice, so I'm unconvinced this proposition explains anything about the algorithm.

I'm not intimately familiar enough with the area to know all relevant related work, but it seems like additional baseline methods could be included to really show that the proposed method is effective. For example, why not compare with AWAC?

The paper is overall a bit difficult to follow. In particular, some key related work (RLPD, CUP) and experimental setup (what does “using average reward to learn offline policies” in 5.2 mean?) is not explained. Figure captions are generally too short; for example, the Fig 4 caption says "distribution of random intent priors, datasets, and behavior cloning policies." The y axis is "Statistics", with a maximum value of 100. Is this a histogram? Figures 6 and 7 are difficult to unpack because the captions are uninformative, and because the text doesn't adequately explain PEX and CUP.

**Questions:**

What is the y axis in Figure 4?

Why is RLPD an oracle? It seems to consistently outperform UBER in some settings.

Is any exploration bonus/epsilon sampling used for TD3 and RLPD? It is surprising that they consistently achieve exactly zero return.

Maybe I’m missing something, but in Figure 4, UBER isn’t clearly learning more diversity? For example, in walker2d, it actually seems like the dataset returns have more mass in the high-reward areas?

Why is the reward-free (vs action-free and reward-free) setting the one we should be interested in? Justifying the problem setting would be helpful.

**Limitations:**

Discussion of when the various assumptions (offline-only vs having online exploration available, reward-free vs reward-free and action-free) are realistic/application would be very helpful.

---

> ### Author Rebuttal · Authors · 2023-08-10
>
> Dear Reviewer,
>
>
> We appreciate the reviewer's valuable feedback.
>
>
> **W1: Concerns about the generosity of the theorems.**
>
> - Theorem 4.2
>
> Recent advanced analysis [1] allows us to refine the suboptimality bound of Theorem 4.2 to be $\tilde{O}(\sqrt{d^2H^3/N})$ without algorithmic adjustments. Then, the performance bound is (nearly) minimax optimal and can not be further improved. Also, it has a nontrivial performance bound. For Mujoco tasks,
> $V_{max}\sim H$, $d\sim 10$, $H\sim 1000$, $N\sim 1e6$, which leads to a guaranteed $68\%$  performance.
>
> Note that the main focus of Theorem 4.2 is not on sample complexity but on **robustness**. Theorem 4.2 allows us to learn an optimal policy for **any** intention $z$ as long as the corresponding policy $\pi^*_z$ is covered by the dataset. This indicates that offline algorithms are robust to reward functions, which allows us to learn diverse behaviors from one *single* dataset.
>
> - Proposition 4.3
>
> We agree with the reviewer that Proposition 4.3 can be coarse. To give a finer analysis of how the random rewards functions can cover the true reward function, we refine our theory and conduct further experiments.
>
> We resort to the random feature theory [2] for the theory part. Specifically, we can show that $\tilde{O}(\sqrt{M})$ random intentions are enough to cover the true intention, where $M$ is the size of the dataset. Please see the general response for more details.
>
> For the empirical part, to show that the set of random intentions does cover the set of true intentions, we calculate the correlation with the true reward and linear projection error for $N=256$ random rewards for each task.
>
> The results are shown in Table 1 in our general response. Random intentions do have a high correlation with the true intention, and a linear combination of random rewards can cover the true reward function.
>
>
> **W2: Additional baselines and comparison with AWAC.**
>
> **A for W2:** As suggested, we compare with the offline unsupervised behavior extraction methods (OPAL [10] and PARROT [11]) and unsupervised data sharing methods (UDS [6]).
>
> The experimental results in Figure 1 in the General Response show that UBER performs better than these baselines in most tasks.
> It is unsurprising because prior methods extract behaviors in a behavior-cloning manner, which lacks diversity and leads to degraded performance for downstream tasks.
>
> **W3: Clear description.**
>
> **A for W3:** We appreciate the detailed and valuable comments.
> In the next revision, we will improve the presentation of the paper by providing a detailed explanation of the experimental setup, other related works (PEX, CUP, and RLPD), and figure captions.
>
> **Q1: Unclear y-axis:**
>
> **A for Q1:** The y-axis represents the unnormalized frequency of each return range. We have updated the visuals to present normalized return distribution for clarity. Please refer to the attached PDF.
>
> **Q2: Why is RLPD an oracle?**
>
> **A for Q2:** RLPD is a SOTA offline-to-online method that **uses the reward information** in the offline dataset, while our method focuses on the unsupervised setting where the reward information is unavailable.
>
>
> **Q3: Exploration bonuses in TD3 and RLPD:**
>
> **A for Q3:** Neither TD3 nor RLPD uses exploration bonuses. However, for environments like Antmaze, common exploration strategies fall short due to the environment's complexity and the sparsity of reward signals. Here, offline datasets are imperative for meaningful results.
>
> **Q4: In Figure 4, UBER isn't clearly learning more diversity?**
>
> **A for Q4:** To provide a clearer view of UBER promoting diversity, we have calculated the entropy of the return distribution, as detailed in Table 2 in our General Response. UBER prominently encourages diversity across all tasks, with the sole exception being walker2d-expert. We hypothesize that expert data dominate the dataset and the task has low sensitivity to reward variations [3]. Also, in this case, the lack of diversity is not a big problem since the optimal behavior is already in the behavior set.
>
>
> **Q5: Why is the reward-free (vs action-free and reward-free) setting the one we should be interested in?**
>
> **A for Q5:**
>
> - Reward-free settings naturally appear in many real-world problems. 1. In real-world problems like robotic tasks and NLP tasks [4,5], reward labels are expensive to get, while action labels are relatively cheap. 2. The setting also appears in data-sharing problems between different tasks [6,7]. We can remove the reward label for other tasks and reuse them as reward-free data for new tasks.
>
>
> - Action labels are usually cheap while being crucial for efficient learning.
>    Their absence hinders our ability to estimate transition models or behaviors, necessitating either additional assumptions [8] or a considerable data volume [9].
>
>
> **L1: Limitations**
>
> **A for L1:** We thank the reviewer for pointing this out, and we will add discussion for the limitation in the updated manuscript.
>
>
> Thanks again for the valuable comments.
> We hope our response has clarified your concerns.
> We are looking forward to further feedback and discussions.
>
> Best,
>
> The Authors
>
> References
>
> [1] Xiong, Wei, et al. "Nearly minimax optimal offline reinforcement learning with linear function approximation: Single-agent mdp and Markov game." arXiv preprint arXiv:2205.15512 (2022).
>
> [2] Rudi, Alessandro, and Lorenzo Rosasco. "Generalization properties of learning with random features." Advances in neural information processing systems 30 (2017).
>
> [3] Li, Anqi, et al. "Survival Instinct in Offline Reinforcement Learning." arXiv preprint arXiv:2306.03286 (2023).
>
> [4] Ouyang, Long, et al. "Training language models to follow instructions with human feedback." Advances in Neural Information Processing Systems 35 (2022): 27730-27744.
>
> [5] Christiano, Paul F., et al. "Deep reinforcement learning from human preferences." Advances in neural information processing systems 30 (2017).

---

> > ### Author Response · Authors · 2023-08-10
> > **Additional Reference**
> >
> > [6] Yu, Tianhe, et al. "How to leverage unlabeled data in offline reinforcement learning." International Conference on Machine Learning. PMLR, 2022.
> >
> > [7] Hu, Hao, et al. "The provable benefits of unsupervised data sharing for offline reinforcement learning." arXiv preprint arXiv:2302.13493 (2023).
> >
> > [8] Torabi, Faraz, Garrett Warnell, and Peter Stone. "Recent advances in imitation learning from observation." arXiv preprint arXiv:1905.13566 (2019).
> >
> > [9] Baker, Bowen, et al. "Video pretraining (vpt): Learning to act by watching unlabeled online videos." Advances in Neural Information Processing Systems 35 (2022): 24639-24654.
> >
> > [10] Ajay et al., OPAL: Offline Primitive Discovery for Accelerating Offline Reinforcement Learning, 2021.
> >
> > [11] Singh et al., Parrot: Data-Driven Behavioral Priors for Reinforcement Learning, 2021.

---

> > ### Comment · Area_Chair_UoAx · 2023-08-19
> >
> > Dear authors,
> >
> > Thank you for submitting your response to the comments.
> >
> > Dear Reviewer XkfL,
> >
> > Were your concerns addressed by the authors?
> >
> > Best,
> >
> > AC

---

> > ### Comment · Reviewer_XkfL · 2023-08-19
> >
> > I appreciate the author's comprehensive response, additional theoretical contributions, and experimental evaluations. With the understanding that the clarity of writing needs improvement (in terms of explaining prior work, experimental results, and more clearly captioning figures with informative captions) before publication, I would be okay with accepting the paper. Therefore I raise my score to 6.

---

> > > ### Author Response · Authors · 2023-08-19
> > > **Thanks for raising the score to 6!**
> > >
> > > We would like to thank the reviewer for raising the score! We really appreciate the valuable comments and suggestions from the reviewer.

---

> ### Author Response · Authors · 2023-08-17
> **Looking forward to further comments!**
>
> Dear reviewer,
>
> We have updated our supplementary experimental results and a more in-depth explanation of UBER. We also updated enhanced theory results. We are wondering if our response and revision have cleared your concerns. We would appreciate it if you could kindly let us know whether you have any other questions. We are looking forward to comments that can further improve our current manuscript. Thanks!
>
> Best regards,
>
> The Authors

---

### Official Review · Reviewer_822o · 2023-07-07

**Soundness:** 3 good
**Presentation:** 3 good
**Contribution:** 3 good
**Rating:** 7
**Confidence:** 3

**Summary:**

The paper studies a setting where there is an offline trajectory dataset with no reward information and the goal is to extract effective behaviors from the offline data such that they can be re-used during a separate online phase to accelerate online learning. To extract effective behaviors from the offline data, the authors propose to use an offline RL algorithm (TD3+BC) to pre-train on the offline dataset with random reward functions, resulting a policy for each random reward function. Then, during the online phase, a new discrete-action policy is initialized and being optimized to select the set of pre-trained policies obtained from the offline phase using a standard online RL (TD3). The paper also provides theoretical argument for why using random reward functions for behavior extraction is sufficient and effective. Empirically, the proposed method is able to outperform existing methods on D4RL AntMaze tasks and Locomotion tasks.

**Strengths:**

- The idea of extracting behavior prior and learning a selection policy online has been explored in some prior works (e.g., [1]). The authors should definitely discuss how this is related to the proposed approach here. Despite that, the use of random reward network for extracting the behaviors is novel (along with theoretical analyses that justify the idea)
- Empirical results (especially on AntMazes) are strong, suggesting that the proposed method is effective at extracting behaviors that sufficient for accelerating online learning.
- The paper is well-written and easy to follow.

[1] Singh, Avi, et al. "Parrot: Data-driven behavioral priors for reinforcement learning." arXiv preprint arXiv:2011.10024 (2020).

**Weaknesses:**

- For any behaviors that are not covered in the offline dataset, the proposed method would not be able to capture them well. If the online task requires new unseen behaviors, the learning might fail completely. This limitation should be addressed/discussed in more details.

**Questions:**

- L190 -- the authors mention that there are also visual tasks but I could not find them in the paper.

**Limitations:**

See the first point in the weakness section.

---

> ### Author Rebuttal · Authors · 2023-08-10
>
> Dear Reviewer,
>
> We appreciate the Reviewer for finding our work novel, effective and well-written. We provide clarification to the points the Reviewer raised as follows.
>
> **S1: Discussion and comparison with previous behavior extraction methods.**
>
> **A for S1:** Previous behavior extraction methods can be divided into two categories:
>
> - **Online Skill Extraction**: Notable methods in this category [1, 2] are proposed in the presence of an online interactive environment. Their reliance on an exploration objective (e.g., information gain) to learn diverse skills makes them less applicable in offline scenarios. Directly leveraging these methods with an offline dataset could introduce significant extrapolation errors.
>
> - **Offline Hierarchical RL**: Techniques that employ offline dataset reuse, such as the ones presented in [3, 4], predominantly utilize behavior cloning for skill extraction over extended temporal scales. While effective when the dataset closely matches the test environment, these methods might not consistently generate diverse behaviors, which are imperative for learning novel tasks.
>
> In contrast, our work focuses on **offline unsupervised RL**, where agents extract diverse behaviors in the reward-free offline dataset.
> We conducted additional experiments to compare with prior offline unsupervised behavior extraction methods, OPAL and PARROT. Specifically,
>
> - We use the VAE model consistent with OPAL to extract behavioral policy and reuse it based on PEX during the online phase. We name this measure OPAL-PEX.
>
> - We use the Flow model consistent with PARROT to extract behavioral policy and reuse it based on PEX during the online phase. We name this measure PARROT-PEX.
>
> - In addition, we set the reward of the dataset to 0 and then learn the offline behavioral policy. Next, reuse it based on PEX during the online phase. We name this measure as UDS-PEX.
>
> The experimental results in Figure~1 in General Response show that UBER performs better than these baselines in most tasks.
> It is unsurprising because prior methods extract behaviors in a behavior-cloning manner, which lacks diversity and leads to degraded performance for downstream tasks.
>
>
> **W1: The method would fail if the online task requires new unseen behaviors.**
>
> **A for W1:**
> It's worth noting that UBER includes a randomly initialized and learnable policy in the behavior set, so it will not completely fail when the online task requires new behaviors. If so, the performance will degrade to pure online learning but not fail. We thank the Reviewer for pointing this out, and we will discuss the limitation in the updated manuscript.
>
>
>
> **Q1: The visual tasks.**
>
> **A for Q1:** These are typos and should be multi-task (i.e., meta-world) rather than visual tasks, where we have multiple datasets and downstream tasks. Please see Appendix B for more details of the experiments on multi-task settings.
>
> Thanks again for your supportive comments and suggestions.
> We sincerely hope that our response has addressed your concerns. Any further feedback and discussions are highly appreciated.
>
>
> Best,
>
> The Authors
>
>
> References
>
> [1] Eysenbach, Benjamin, et al. "Diversity is all you need: Learning skills without a reward function." arXiv preprint arXiv:1802.06070 (2018).
>
> [2] Sharma, Archit, et al. "Dynamics-aware unsupervised discovery of skills." arXiv preprint arXiv:1907.01657 (2019).
>
> [3] Ajay, Anurag, et al. "Opal: Offline primitive discovery for accelerating offline reinforcement learning." arXiv preprint arXiv:2010.13611 (2020).
>
> [4] Singh, Avi, et al. "Parrot: Data-driven behavioral priors for reinforcement learning." arXiv preprint arXiv:2011.10024 (2020).

---

> > ### Comment · Area_Chair_UoAx · 2023-08-19
> >
> > Dear authors,
> >
> > Thank you for your taking the time to respond to the comments.
> >
> > Dear Reviewer 822o,
> >
> > After reading the authors' response, do you have any additional thoughts?
> >
> > Best,
> >
> > AC

---

### Official Review · Reviewer_VsN9 · 2023-07-11

**Soundness:** 2 fair
**Presentation:** 3 good
**Contribution:** 2 fair
**Rating:** 6
**Confidence:** 4

**Summary:**

This paper tackles the problem of unsupervised behavior extraction from reward-free offline data. The main idea is to pre-train multiple policies with random rewards. UBER consists of two phases. It first trains $N$ ($100$ or $256$) policies with random rewards with an offline RL algorithm (TD3+BC), and in the subsequent online phase, it continues training the N policies plus a newly initialized policy with a soft policy selector based on the Q functions. The authors show that the behaviors learned by UBER are helpful to solve downstream tasks in standard offline RL benchmarks.

**Strengths:**

- The proposed method seems novel to me and is relatively easy to implement.
- Despite the simplicity, the behaviors learned by random rewards seem helpful in various downstream tasks in the D4RL benchmark.
- The paper contains several analyses including ablation studies.
- The paper is well-written and easy to understand.

**Weaknesses:**

- The theoretical results do not seem to justify the use of **random** rewards. Theorem 4.2 states a general convergence result in offline RL, and Theorem 4.3 states that if two reward functions are similar, the corresponding optimal value functions are also similar. I'm not convinced how these theorems support the effectiveness of *random* reward functions. Figure 8 in Appendix A.3 states that if random reward functions sufficiently cover the reward function space, any task reward function can be approximated by the closest random reward function. However, it is unclear as to how many random reward functions are needed to enjoy this benefit. For example, we may need exponentially many random reward functions (i.e., $O((1/\epsilon)^{|S||A|})$) to cover the entire reward function space. I would have expected a complexity analysis similar to Theorem 3.1 in Chen et al. [1].
- The reason why random rewards lead to diverse behaviors in Figure 1 may heavily depend on the (strict) early termination condition in Hopper (and the same for Walker2d). How do the behaviors from random reward functions look like in HalfCheetah and AntMaze, which do not have early termination conditions? Could the authors provide videos and/or plots similar to Figure 4 in these environments?
- The paper lacks discussions/comparisons with prior offline unsupervised behavior (or behavioral prior) extraction methods (e.g., OPAL [2], SPiRL [3], and PARROT [4]) and prior unsupervised data sharing methods (e.g., UDS [5] and PDS [6]).

Typos and minor comments
- L117: Missing citation.
- OPAL [2] is cited but not mentioned in the manuscript.
- What does PEX stand for?

[1] Chen et al., Self-Supervised Reinforcement Learning that Transfers using Random Features, 2023.

[2] Ajay et al., OPAL: Offline Primitive Discovery for Accelerating Offline Reinforcement Learning, 2021.

[3] Pertsch et al., Accelerating Reinforcement Learning with Learned Skill Priors, 2020.

[4] Singh et al., Parrot: Data-Driven Behavioral Priors for Reinforcement Learning, 2021.

[5] Yu et al., How to Leverage Unlabeled Data in Offline Reinforcement Learning, 2022.

[6] Hu et al., The Provable Benefit of Unsupervised Data Sharing for Offline Reinforcement Learning, 2023.

**Questions:**

Based on the weaknesses section above, my two biggest questions are:
- How does UBER compare to previous unsupervised behavior extraction and/or data-sharing methods? I do not expect comparisons with all the above methods, but it would be nicer if the authors could provide empirical comparisons with some of the methods in these categories (or discussions about why UBER is very different from them).
- Why is using **random** reward functions a good idea when extracting behaviors? If this is for purely empirical reasons, I'm fine with that (though it would have required more thorough empirical evaluations in diverse environments), but at least the theorems in the paper do not seem to provide an answer to this important question.

**Limitations:**

The paper lacks discussions about the limitations of UBER. One limitation I can imagine is that this random reward strategy may fail in more complex environments and thus may not be scalable (though addressing this limitation may be out of the scope of this work and it does not affect my score).

---

> ### Author Rebuttal · Authors · 2023-08-10
>
> Dear Reviewer,
>
> Thank you for your constructive feedback. We have provided additional experimental results and explanations to address your concerns, and we hope the following clarifications shed light on the raised points.
>
> **W1: The theoretical results do not seem to justify the use of random rewards.**
>
> **A for W1:**
>
> **(Theorem 4.2)**
> The key focus of Theorem 4.2 is that the offline performance is robust to intention $z$, as long as the dataset covers the corresponding policy $\pi^*_z$.
> Concurrent work [7] also finds the robustness of the reward function for pessimism algorithms, which aligns with our findings.
> This robust result allows us to learn diverse behaviors from a *single* dataset, while prior works either learn one policy from one dataset [5,6] or resort to online interactive environments [8].
>
> **(Theorem 4.3)**
> We agree with the Reviewer that Theorem 4.3 can be coarse in showing coverage, and we thank the Reviewer for the reference.
> We do not need to cover the whole state-action space but only the dataset distribution. Then, we can use the random feature theory for random intention coverage. Specifically, we can show that $\tilde{O}(\sqrt{M})$ random intentions are enough to cover the true intention, where $M$ is the size of the dataset. Please see the general response for more details. This requires ~1000 random intentions for the dataset of size $M=1e6$, which aligns well with our practice with $N=256$.
>
> **W2: How do the behaviors from random reward functions look like in HalfCheetah and AntMaze?**
>
> **A for W2:** We provide similar plots for Halfcheetah and Antmaze tasks in the attached PDF file as Figure 4 in our submission.
> We can see that UBER is encouraging diverse behaviors regardless of strict terminating conditions. This means that the diversity does not (only) come from terminating at different timesteps but from diverse intentions.
>
> **W3 \& Q1: Lack discussions/comparisons with previous unsupervised behavior extraction and data-sharing method.**
>
> **A for W3 \& Q1:** As suggested, we compare with the offline unsupervised behavior extraction methods and unsupervised data sharing methods, including OPAL, PARROT, and UDS. The experimental results in Figure~1 in General Response show that UBER performs better than these baselines in most tasks.
> The result is not surprising because prior methods extract behaviors in a behavior cloning manner, which lacks diversity and leads to degraded performance for downstream tasks, especially when the downstream tasks differ from the dataset.
>
> **Q2: Why is using random reward functions a good idea when extracting behaviors?**
>
> **A for Q2:**
>
> The motivation for using random rewards is to provide a simple way to extract *diverse* yet useful behaviors. Previous online behavior extraction methods use an explorative objective (e.g., information gain) to acquire diverse behaviors, which is not applicable in the offline setting since it will lead to extrapolation errors and value explosion. Previous offline methods [1,2] learn temporal-extended skills in a behavior-cloning manner, which lacks diversity and leads to degraded performance for downstream tasks.
>
> Then, we justify the use of random rewards empirically and theoretically:
>
> Empirically, using random reward functions can be justified from two aspects: the coverage of the true reward and the diversity of the behavior set with strong empirical performance.
>
> **1. Coverage:**
>
>  - To show that the set of random intentions does cover the set of true intentions, we calculate the correlation with the true reward as well as linear projection error for each task.
>
>  - The experimental results in Table 1 in General Response show that random intentions do have a high correlation with the true intention, and a linear combination of random rewards can well cover the true reward function. Note that we are using the random reward **functions** with $(s,a)$ as input rather than completely random. The latter will lead to near zero correlation with the true intention and about $40\%$ projection error.
>
> **2. Diversity:**
>
> - Figure 4 in our submission clearly shows that UBER is encouraging diversity since it has a wider span of distributions over the returns. To show this more explicitly, we further calculate the entropy of the return distribution, as shown in Table 2 in the General Response.
>
> Theoretically, using random reward functions is justified from two perspectives: robustness and coverage.
>
> **1. Robustness:**
>
> - For robustness, as stated in the answer of W1, Theorem 4.2 in our work and concurrent work [3] both show that pessimism leads to robustness over rewards. This allows us to smooth out the small fluctuations in the reward function and enables learning diverse behaviors from one dataset.
>
> **2. Coverage:**
>
> - For coverage, as stated in our answer to W1, we can show that a reasonable number of random reward functions is enough to cover the true reward. Note that we do not need to cover the whole state-action space, but the support of the dataset and $\tilde{O}(\sqrt{M})$ random functions is sufficient, where $M$ is the size of the dataset.
>
> **L1:  The applicability of the proposed method for complex rewards.**
>
> **A for L1:** We thank the Reviewer for pointing this out, and we will discuss the limitation in the updated manuscript.
>
> Thanks again for the detailed and valuable comments. We sincerely hope our response can clear your concerns and look forward to more discussions.
>
> Best,
>
> The Authors
>
> [1] Ajay, Anurag, et al. "Opal: Offline primitive discovery for accelerating offline reinforcement learning." arXiv preprint arXiv:2010.13611 (2020).
>
> [2] Singh, Avi, et al. "Parrot: Data-driven behavioral priors for reinforcement learning." arXiv preprint arXiv:2011.10024 (2020).
>
> [3] LI, Anqi, et al. Survival Instinct in Offline Reinforcement Learning. arXiv preprint arXiv:2306.03286, 2023.

---

> > ### Comment · Reviewer_VsN9 · 2023-08-11
> >
> > Thanks for the detailed response! I believe the new theorem and the qualitative results on HalfCheetah and AntMaze do improve the quality of the paper, and I raised my score from 4 to 6.

---

> > > ### Author Response · Authors · 2023-08-11
> > > **Thanks for raising the score to 6!**
> > >
> > > We would like to thank the reviewer for raising the score! We really appreciate the valuable comments and suggestions from the reviewer.

---

### Official Review · Reviewer_wm6d · 2023-07-12

**Soundness:** 1 poor
**Presentation:** 3 good
**Contribution:** 2 fair
**Rating:** 4
**Confidence:** 4

**Summary:**

The authors propose unsupervised behavior extraction via random intent priors (UBER), an unsupervised method for extracting and learning behaviors from an offline dataset for downstream tasks. Assuming the situations where there are no reward labels for the transitions in the offline dataset, they suggest using a set of random *intentions* and thus intention-induced random reward functions to train a set of policies. For the online downstream task learning, they employ the policy expansion scheme and reuse the learned policies along with a newly learning policy based on the output of the critic. The authors also provide theoretical results on the existence of the corresponding intention given an arbitrary behavior, a bound on the suboptimality of the policy that is trained with some intention in a linear MDP setting, and the robustness of the random reward functions. Empirically, they present the comparison of performance and analyses in MuJoCo, AntMaze, and Meta-World environments.

**Strengths:**

- The proposed method shows good empirical performance in general compared to the baselines on the downstream tasks in MuJoCo, AntMaze, and Meta-World.
- The manuscript is easy to follow and clearly structured. It is also equipped with appropriate conceptual figures, which can help readers' understanding.
- Learning from unlabeled offline behavior data is an important topic in RL, given that much more unlabeled behavior data is available compared to labeled data in the real world.

**Weaknesses:**

- My current major concern is that the use of random intentions doesn't seem well-justified (and motivated) to me given the current state of this submission. While Fig.4 suggests that UBER's learned behaviors cover the distribution of *behaviors* from the original offline dataset in terms of their resulting returns, it may not necessarily mean that the random intentions or the random reward functions used for the training cover/match (or are highly correlated with) the true intentions. There is a fair possibility that the pessimism in the offline RL training is playing an important role in matching the offline dataset's behaviors, which is also suggested by a concurrent work [1]. I imagine that the whole *intention* space might be too large to be covered with $N=100$ or $N=256$, but it may not matter for the online phase in practice as the policy selection is performed at every time step.
- Some writing or editing issues
  - A missing reference at L117.
  - The definition (or notations) of the loss functions in Algorithms 1 and 2.
  - I believe calling RLPD an *oracle* because of the use of the offline dataset with the true rewards for its training can be misleading.
  - Behavior size $N$ (from the main manuscript) vs Random reward dim $n$ (from the appendix)?

[1] LI, Anqi, et al. Survival Instinct in Offline Reinforcement Learning. *arXiv preprint arXiv:2306.03286*, 2023.

**Questions:**

- Could you provide analyses to back up the claim that the set of random *intentions* is covering the set of true intentions?
- Regardless of my concern, I suggest the authors to check out the concurrent work [1].

**Limitations:**

Please take a look at the Weaknesses and Questions sections above.

---

> ### Author Rebuttal · Authors · 2023-08-10
>
> Dear Reviewer,
>
> Thank you for your constructive feedback. We've provided additional experimental results and explanations to address your concerns, and we hope the following clarifications shed light on the raised points.
>
> **Q1 \& W1.1: The motivation and justification of using random intentions and their alignment with true intentions.**
>
> **Response for Q1 \& W1.1:** We employ random rewards as a simple yet effective mechanism to distill diverse and useful behaviors from offline data.
>
> In online settings, behavior extraction typically uses exploration objectives, like information gain, to produce diverse behaviors. Such objectives could be better-suited for offline settings, as they can cause extrapolation errors and result in value explosion. On the other hand, existing offline methods learn temporally-extended skills using behavior cloning, but they may not produce enough behavioral diversity. Lack of diversity leads to suboptimal performance, especially when the downstream task differs from the provided dataset.
>
>
> To substantiate our claim that random intentions effectively cover true intentions, we highlight the following:
>
> **(Experiments)** We calculate the correlation of $N=256$ random rewards with the true reward and measure the linear projection error. Our results (Table 1 in General Response) indicate that random intentions can have a high correlation with true intentions, and linear combinations of random rewards can approximate the true reward function quite well. Note that we use random reward functions (represented by neural networks) based on $(s, a)$ inputs rather than entirely random ones, which have a near-zero correlation with the true reward and $40\%$ projection error.
>
>
> **(Theory)** Drawing parallels to the random features theory, we consider each random reward function as a unique dimension of a random feature, enabling us to leverage the random feature theory to affirm our random intention coverage. A detailed elaboration is available in the general response.
>
> **Q2 \& W1.2: Comparing our findings with concurrent work [1] and the role of pessimism in aligning offline behaviors.**
>
> **Response for Q2 \& W1.2:** Concurrent work [1] indeed resonates with some of our observations, especially Theorem 4.2. While the concurrent study emphasizes the robustness of offline algorithms to reward variations, our findings suggest that they can adeptly learn tasks with various intentions $z$ as long as the optimal policy is represented in the dataset. So, Theorem 4.2 is also a kind of "robustness over the reward" statement for pessimistic offline algorithms. This robustness helps smooth the random reward function and makes an implicit trade-off between usefulness and diversity. This is also observed empirically since a reasonable number of behaviors have nontrivial or near-optimal performance.
>
>
> However, this robustness alone doesn't encapsulate the efficacy of our approach. For instance, behaviors learned with random intentions are diverse and have significant entropy in their return distributions (Figure 4). It would be contradictory if the reward function's robustness were the only driving factor. Additionally, our method consistently outperforms behavior cloning techniques, even when behavior cloning leads to near-optimal policy (e.g., in expert datasets). This further underscores the importance of behavioral diversity over mere reward robustness.
>
> **W2: Concerns about the quality of writing.**
>
> **Response for W2:** Thank you for pointing these out. We've rectified the issues in our revised manuscript to enhance clarity and readability.
>
> We deeply appreciate your thorough feedback and sincerely hope our clarifications address your concerns. Any further feedback and discussions are highly appreciated.
>
> Best regards,
>
> The Authors

---

> ### Author Response · Authors · 2023-08-17
> **Looking forward to further comments!**
>
> Dear reviewer,
>
> We have updated our supplementary experimental results and a more in-depth explanation of UBER. We also updated enhanced theory results. We are wondering if our response and revision have cleared your concerns. We would appreciate it if you could kindly let us know whether you have any other questions. We are looking forward to comments that can further improve our current manuscript. Thanks!
>
> Best regards,
>
> The Authors

---

> > ### Comment · Area_Chair_UoAx · 2023-08-19
> >
> > Dear authors,
> >
> > Thank you for submitting your response to the comments.
> >
> > Dear reviewer wm6d,
> >
> > Were your concerns addressed by the authors?
> >
> > Best,
> >
> > AC

---

### Official Review · Reviewer_MUZS · 2023-07-17

**Soundness:** 3 good
**Presentation:** 3 good
**Contribution:** 3 good
**Rating:** 5
**Confidence:** 4

**Summary:**

The paper studies the usage of unsupervised (reward-free) data to help RL, which could extract useful behaviors from offline reward-free datasets. The proposed method is called UBER, which generates random intent priors and trains the agents based on them. The procedure generates diverse behaviors which in turn helps RL learn in a sample efficient way.

**Strengths:**

+ The proposed method is clearly motivated and presented. The illustration figure and the algorithm description are easy to read;

+ The proposed method has theoretical support and proofs;

+ In the empirical experiments, the proposed method is much more sample efficient than baselines due to it learns useful behaviors to help the actual task.

**Weaknesses:**

I feel that the "unsupervised RL to learn a set of diverse behavior so that when the actual reward function applies, the RL agents can learn much faster" idea appeared in the literature. And the paper seems innovative in the setting part where the unsupervised RL happens in an offline RL fashion.

**Questions:**

Can the authors justify the novelty part when it compares to the unsupervised RL literature?

---

> ### Author Rebuttal · Authors · 2023-08-10
>
> Dear Reviewer,
>
> Thank you for taking the time to review our manuscript and for providing insightful feedback. We appreciate the opportunity to clarify our contributions and address your concerns.
>
> **W1 \& Q1: Justify the novelty part when it compares to the unsupervised RL literature.**
>
> **A for W1 \& Q1:**
>
> The novelty of our work comes from three perspectives: a new setting, a simple yet effective algorithm, and its theoretical justification.
>
> First, we propose a new setting: leveraging unsupervised or reward-free offline datasets to accelerate online learning for novel tasks. This helps reduce exploration costs and enable data reuse from other tasks (e.g., we can discard reward labels in the dataset from other tasks). It also enables learning from large datasets, which has the potential for behavior emergence, which is impossible for online RL due to its low efficiency. Traditional unsupervised RL methods require an online interaction environment [1,2] or oracle reward functions [3,4].
>
> Secondly, we propose a simple yet effective method using random intentions to extract diverse and useful behaviors.
> Previous work on online skill extraction relies on an exploration objective (like information gain) to derive diverse skills, which makes them less applicable in offline scenarios. Directly leveraging these methods with an offline dataset could introduce significant extrapolation errors. Previous techniques that employ offline dataset reuse, such as the ones presented in [3,4], predominantly utilize behavior cloning for skill extraction over extended temporal scales. While effective when the dataset closely matches the test environment, these methods might not consistently generate diverse behaviors, which are imperative for learning novel tasks.
>
> Thirdly, we provide a theoretical argument for why using random reward functions for behavior extraction is sufficient and effective. We use offline RL theory and random feature theory to show the robustness and coverability in the reward space of random rewards, which justifies using random intentions.
>
>
> We genuinely hope that our explanation clarifies the distinctiveness and potential of our work. We are eager to engage in further discourse if needed.
>
> Sincerely,
>
> The Authors
>
>
> References
>
> [1] Eysenbach, Benjamin, et al. "Diversity is all you need: Learning skills without a reward function." arXiv preprint arXiv:1802.06070 (2018).
>
> [2] Sharma, Archit, et al. "Dynamics-aware unsupervised discovery of skills." arXiv preprint arXiv:1907.01657 (2019).
>
> [3] Ajay, Anurag, et al. "Opal: Offline primitive discovery for accelerating offline reinforcement learning." arXiv preprint arXiv:2010.13611 (2020).
>
> [4] Singh, Avi, et al. "Parrot: Data-driven behavioral priors for reinforcement learning." arXiv preprint arXiv:2011.10024 (2020).

---

> > ### Comment · Reviewer_MUZS · 2023-08-14
> >
> > Thanks for your rebuttal. My concerns have been addressed.

---

> > > ### Author Response · Authors · 2023-08-17
> > >
> > > Dear Reviewer,
> > >
> > > Thanks for your time and effort in the review process, and we truly appreciate your constructive feedback, which allowed us to improve our work.
> > >
> > > We are delighted to know that our rebuttal has resolved your concerns. However, it appears that the overall score has remained unchanged. We kindly request that you re-evaluate the score in light of the resolved concerns, if possible.
> > >
> > > We understand that the review process can be quite demanding, and details can sometimes be overlooked. We sincerely hope our note serves only as a gentle reminder and not as a presumption.
> > >
> > > Warm regards,
> > >
> > > The Authors

---

### Author Rebuttal · Authors · 2023-08-10

Dear Reviewers,


We thank all the reviewers for their constructive feedback and valuable insights. We are encouraged to learn that many found our work "novel," "interesting," "fairly promising," and "well-written." We genuinely appreciate these positive remarks.

We acknowledge the concerns raised about the lack of comprehensive theoretical results, experimental data, and the clarity of our algorithm's explanation. In response:

- We have included enhanced theoretical results.
- Our supplementary section now showcases expanded experimental results, comparing UBRE with additional unsupervised behavior extraction and data-sharing techniques.
- We've also provided a more in-depth explanation of UBER tailored to each reviewer's feedback.

We sincerely hope that these updates and clarifications will address the reviewers' concerns. More discussions and suggestions for further improving the paper are welcomed!

### New Theoretical Results

To give a stronger theoretical guarantee for using random intentions, we resort to random feature theory for a better characterization of the coverage property of random functions. Especially, we have the following theorem.

**Theorem.** Assume the reward function admits a RKHS represention $\psi(s,a)$ with $|\psi(s,a)|\leq \kappa$ almost surely. Then with $N=c_0 \sqrt{M}\log(18\sqrt{M}\kappa^2/\delta)$ random reward functions, the linear combination of the set of random reward functions can approximate the true reward function with error

$$
\epsilon \leq c_1 \log^2(18/\delta)/\sqrt{M},
$$

with probability $1-\delta$.

The key of the proof is noticing that a random reward function can be seen as one random feature for linear regression. We will update the manuscript for the theorem and its full proof. The theorem above allows us to use $N=\tilde{O}(\sqrt{M})$ random rewards to well cover the true reward function, where $M$ is the size of the dataset.
We hope the theorem above can solve the concerns on the theoretical justification of using random intentions.

### Additional Experiments:

- Correlation ratio and linear projection error on various tasks:

| Task | Max Correlation | Min Correlation | Projection Error |
|-----|------|------|-----|
|hopper-medium-v0 | 0.569 |-0.568|0.016 |
|hopper-medium-expert-v0 |0.498 |-0.540|0.015 |
|hopper-expert-v0 | 0.423 |-0.415 | 0.011 |
|halfcheetah-medium-v0 |0.569 |-0.568|0.016 |
|halfcheetah-medium-expert-v0 |0.605 |-0.589|0.047 |
|halfcheetah-expert-v0 |0.370|-0.461|0.021 |
|walker2d-medium-v0 |0.475 |-0.582|0.046 |
|walker2d-medium-expert-v0 |0.495 |-0.472|0.042 |
|walker2d-expert-v0 |0.358 |-0.503|0.019 |

Table 1. Minimum and maximum correlation and linear projection error for the true reward function using random reward functions on various tasks. The projection error $\epsilon$ is defined as $\epsilon = ||r-\hat{r}||/||r||$ , where $\hat{r}$ is the best approximation using linear combination of random rewards.

- Distribution of random intent priors, datasets, and behavior cloning policies:

| Method\Dataset | halfcheetah-medium | halfcheetah-medium-expert | halfcheetah-expert | hopper-medium          | hopper-medium-expert | hopper-expert |
|----------------|--------------------|---------------------------|--------------------|------------------------|----------------------|---------------|
| BC             | 0                  | 0                         | 0                  | 0                      | 0                    | 0             |
| Dataset        | 1.21               | 1.47                      | 0.86               | 0.71                   | 1.97                 | 0.82          |
| UBER           | 2.01               | 1.92                      | 2.69               | 2.45                   | 2.32                 | 1.11          |

| Method\Dataset | walker2d-medium    | walker2d-medium-expert    | walker2d-expert    | antmaze-medium-diverse | antmaze-medium-play  |               |
|----------------|--------------------|---------------------------|--------------------|------------------------|----------------------|---------------|
| BC             | 0                  | 0                         | 0                  | 0                      | 0                    |               |
| Dataset        | 1.44               | 1.88                      | 2.43               | 0.63                   | 0.29                 |               |
| UBER           | 2.67               | 2.61                      | 0.41               | 1.77                   | 1.69                 |               |

Table 2. The entropy of the return distribution for each method on different tasks.

---

### Decision · Program_Chairs · 2023-09-21

**Decision:**

Accept (poster)

**Comment:**

The paper introduces a method called UBER, which addresses the challenge of learning useful behaviors from offline data lacking reward labels. UBER achieves this by pre-training multiple policies with random reward functions in an offline phase. In the subsequent online phase, it further optimizes these policies alongside a newly initialized policy. The authors demonstrate that UBER's learned behaviors are effective for downstream tasks in various environments.

**strengths**

* Extensive experiments
* Simplicity of the proposed method
* Theoretical analysis on connection between using random rewards and extracting behaviors

**weaknesses/suggestions**

* The clarity of writing needs improvement (in terms of explaining prior work, experimental results, and more clearly captioning figures with informative captions)
* The applicability of the proposed method for complex rewards

I think the paper makes a nice contribution that the community will find valuable. However, I encourage the authors to think carefully about how to reflect the comments or resolve the questions from reviewers in the camera ready version.